# MARVEL: ACCELERATING SAFE ONLINE REINFORCEMENT LEARNING WITH FINETUNED OFFLINE POLICY

## ABSTRACT

The high costs and risks involved in extensive environment interactions hinder the practical application of current online safe reinforcement learning (RL) methods. While offline safe RL addresses this by learning policies from static datasets, the performance therein is usually limited due to reliance on data quality and challenges with out-of-distribution (OOD) actions. Inspired by recent successes in offline-to-online (O2O) RL, it is crucial to explore whether offline safe RL can be leveraged to facilitate faster and safer online policy learning, a direction that has yet to be fully investigated. To fill this gap, we first demonstrate that naively applying existing O2O algorithms from standard RL would not work well in the safe RL setting due to two unique challenges: *erroneous Q-estimations*, resulted from offline-online objective mismatch and offline cost sparsity, and *Lagrangian mismatch*, resulted from difficulties in aligning Lagrange multipliers between offline and online policies. To address these challenges, we introduce **Marvel**, a novel framework for O2O safe RL, comprising two key components that work in concert: *Value Pre-Alignment* to align the Q-functions with the underlying truth before online learning, and *Adaptive PID Control* to effectively adjust the Lagrange multipliers during online finetuning. Extensive experiments demonstrate that Marvel significantly outperforms existing baselines in both reward maximization and safety constraint satisfaction. By introducing the first policy-finetuning based framework for O2O safe RL, which is compatible with many offline and online safe RL methods, our work has the great potential to advance the field towards more efficient and practical safe RL solutions.

## 1 INTRODUCTION

Safe reinforcement learning (safe RL) (Gu et al., 2022; García & Fernández, 2015), prioritizes not only the maximization of rewards but also the adherence to specific safety constraints, enhancing its applicability in real-world scenarios. For example, an autonomous vehicle must reach its destination without exceeding a preset fuel limit. However, solving safe online RL from scratch in fields such as robotics (Brunke et al., 2022; Kiran et al., 2021), and healthcare (Yu et al., 2021; Qayyum et al., 2020) is often prohibitive, due to substantial risks and costs caused by the need for extensive interactions with the environment. To address this, offline safe RL (Achiam et al., 2017; Zheng et al., 2024; Ray et al., 2019) has been introduced, enabling the derivation of safe policies from a static dataset (Liu et al., 2023b) without the need for real-time environmental interaction. Nonetheless, offline safe RL faces its own set of limitations: it typically shows limited performance (Ghosh et al., 2022), heavily relies on the quality of the offline dataset, and suffers from the impact of out-of-distribution (OOD) actions, restricting its effectiveness across varying scenarios.

The pretraining-and-finetuning paradigm is a well-established strategy in the fields of computer vision and natural language processing, for enabling fast and sample-efficient online learning based on offline pretrained models, particularly with the recent advances in large language models. Following a similar line, offline-to-online reinforcement learning (O2O RL) in the unconstrained case (Nair et al., 2020; Wang et al., 2024; Zhang et al., 2023b) and imitation learning (Yue et al., 2024; Ross et al., 2011) has recently gained prominence. These approaches utilize policies (including Q-functions) derived from offline RL or offline imitation learning, along with offline datasets, to expedite the process of online finetuning. This strategy effectively avoids the extensive environmental interactions required in training policies from scratch. Thus motivated, a key insight is that leveraging the pretraining-and-finetuning paradigm can also potentially facilitate more efficient and

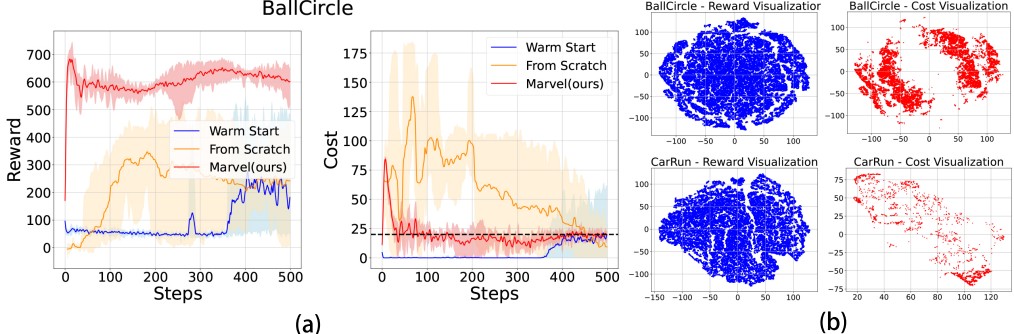

Figure 1: The "steps" on the x-axis represent the number of policy gradient updates (i.e., optimizer updates). For each gradient update, the agent interacts with the environment for 3 episodes. This convention is followed in the subsequent figures. **In (a)**, we evaluate these methods in the BallCircle environment from Bullet Safety Gym (Gronauer, 2022), setting the cost limit to 20. As shown, although "Warm Start" begins with a reasonably good initial policy, it performs poorly and overly conservatively, even worse than "From Scratch" in which the policy and Q-functions are initialized randomly. This result suggests that directly finetuning the pretrained policy and Q-functions may actually hinder online learning. In contrast, "Marvel" achieves impressive results, finding a policy with much higher return in just a few online steps while adhering to the cost limit. **In (b)**, t-SNE visualization of state vectors in the environment, reduced to 2D space. Each point represents a state, with rewards uniformly distributed across the space, while costs are sparse, appearing as isolated points or clusters, reflecting their limited association with states.

practical online safe RL, which however has not been fully explored in the literature. To fill this gap, we seek to answer the following question:

*Can we design an effective offline-to-online approach for safe RL to address the limitations of both online safe RL and offline safe RL, thereby enabling fast online safe policy learning?*

However, achieving this is highly nontrivial, and simply applying existing O2O algorithms in conventional RL would not work well here due to unique challenges in safe RL. In Fig. 1 (a), 'Warm Start' refers to using the offline pretrained policy and Q-networks directly initialize an online safe RL algorithm. 'From Scratch' refers to purely online safe RL training. As illustrated, directly finetuning the offline pretrained policy and Q-functions by using standard online safe RL often results in suboptimal performance and, in some cases, complete training failures.

The reasons behind this phenomenon are as follows: a) *Erroneous Q-estimations resulted from objective mismatch and offline cost sparsity.* In order to avoid explorations beyond the offline data and reduce the extrapolation errors, offline safe RL algorithms typically introduce additional regularizations in the objective function to push up the cost estimates of OOD actions, e.g., VOCE (Guan et al., 2024) and CPQ (Xu et al., 2022), leading to a different overall objective from standard online safe RL. More critically, the majority of state-actions in offline datasets for safe RL usually are safe with zero cost (Fig. 1 (b)), resulting in a pretrained cost Q-function that predicts extremely low cost for most in-distribution (IND) state-actions. By erroneously giving high values for OOD state-actions and low values for IND state-actions, the pretrained cost Q-function will conservatively force the online finetuning to stay in the state-action space similar to offline dataset and be reluctant to explore (e.g., cost of "Warm Start" in Fig. 1). b) *Mismatch of Lagrange multipliers.* Many online safe RL algorithms (Stooke et al., 2020; Chow et al., 2018a; Achiam et al., 2017) solve the constrained optimization problem based on the primal-dual approach, which requires a synchronous updating of Lagrange multipliers. Nonetheless, initial values for these multipliers that are matching with the offline policies cannot be obtained from offline safe RL precisely, such that using traditional dual ascent methods to update the Lagrange multipliers may result in slow learning during the online phase even with accurate estimated Q-functions, ultimately degrading the performance of the learned policy. In this work, we seek to design an effective O2O framework for safe RL by addressing these two challenges above.

The main contribution of this work lies in the development of the **warM-stArt safe Reinforcement learning with Value prE-aLignment (Marvel)** framework, which includes two key components: Value Pre-alignment (VPA) and Adaptive PID Control (aPID). More specifically, VPA adjusts the pretrained Q-functions by re-evaluating the offline policy before online learning based on the offline data only, so as to align the distribution of estimated Q-values with that of true Q-values under

the online learning objective for the offline policy. On one hand, by optimistically estimating rewards and pessimistically estimating costs, VPA promotes active exploration during online learning while maintaining the cost below the limit; on the other hand, the active exploration of high-reward state-actions inevitably increases the risk of exploring high-cost state-actions, amplifying the demand of appropriate Langrange multipliers in online finetuning to penalize the cost violations. To jointly handle this risk and the multiplier mismatch problem, instead of directly finding the best initial multipliers, we take an alternative route by seeking to quickly adapt the multipliers to the right values. Particularly, We introduce aPID, an adaptive PID control mechanism that adjusts the Lagrange multipliers based on cost violations, where PID (Proportional-Integral-Derivative) control is a widely used feedback control technique that combines proportional, integral, and derivative components to minimize errors effectively. This approach can quickly stabilize the online finetuning compared to standard dual ascent-based approaches in online safe RL. Extensive experimental results demonstrate the superior performance of our framework over multiple baseline methods on different benchmarks, i.e., Marvel can quickly find a safe policy with the best reward by using only a few online interactions. To the best of our knowledge, Marvel is the first framework that finetunes pretrained offline policy to facilitate fast online learning for safe RL. More importantly, by only leveraging pretrained offline policy/Q-functions and controlling the Lagrange multipliers update, Marvel is compatible with and ready to plug in a lot of state-of-the-art (SOTA) offline and online safe RL approaches.

## 2 PRELIMINARIES

**Constrained Markov Decision Process.** We consider a standard constrained Markov Decision Process (CMDP) (Sutton, 2018; Altman, 2021), defined by a tuple $(S, A, T, R, C, \gamma, \eta, c_{th})$. Here $S \subseteq \mathbb{R}^n$ represents the state space, $A \subseteq \mathbb{R}^m$ denotes the action space, $T : S \times A \times S \to [0, 1]$ is the transition probability function, $R : S \times A \to [0, R_{\max}]$ is the reward function, and $C : S \times A \to [0, C_{\max}]$ is the cost function. $\gamma \in [0, 1]$ is the discount factor, $\eta$ represents the initial state distribution, and $c_{th}$ is the cost threshold that sets the limit on cumulative costs for the policy. A policy $\pi : S \to \mathcal{P}(A)$ is a mapping from states to a probability distribution over actions, where $\pi(a|s)$ denotes the probability of selecting action $a$ in state $s$. In this work, we consider parameterized policies $\pi_\theta$, where $\theta$ denotes the parameters of the policy, typically represented by neural networks in deep RL. Given a policy $\pi$, its cumulative reward under policy $\pi$ is defined as $R(\pi) = \mathbb{E}_{\tau \sim \pi} \left[ \sum_{t=0}^{\infty} \gamma^t r(s_t, a_t) \right]$, where $\tau = (s_0, a_0, s_1, a_1, \dots)$ is a trajectory induced by policy $\pi$, and the expectation is taken over the distribution of trajectories. Similarly, its cumulative cost is defined as $C(\pi) = \mathbb{E}_{\tau \sim \pi} \left[ \sum_{t=0}^{\infty} \gamma^t c(s_t, a_t) \right]$. The Q-function, for a given policy $\pi$, is defined as the expected cumulative reward starting from a state-action pair $(s, a)$ and thereafter following policy $\pi$: $Q^\pi(s, a) = \mathbb{E}_{\tau \sim \pi} \left[ \sum_{t=0}^{\infty} \gamma^t r(s_t, a_t) \mid s_0 = s, a_0 = a \right]$. Similarly, the cost Q-function $Q_c^\pi(s, a)$ is defined as the expected cumulative cost starting from the same state-action pair $(s, a)$ and thereafter following policy $\pi$: $Q_c^\pi(s, a) = \mathbb{E}_{\tau \sim \pi} \left[ \sum_{t=0}^{\infty} \gamma^t c(s_t, a_t) \mid s_0 = s, a_0 = a \right]$. In the context of CMDP, the goal is to find an optimal policy $\pi^*$ that maximizes the cumulative reward $R(\pi)$, subject to the constraint that the cumulative cost $C(\pi)$ does not exceed a predefined threshold $c_{th}$. This can be formulated as the following constrained optimization problem:

$$\max_\pi R(\pi), \quad \text{s.t.} \quad C(\pi) \leq c_{th}. \tag{1}$$

To solve this, a common approach is to apply the Lagrangian relaxation method (Ray et al., 2019), where a Lagrange multiplier $\lambda$ is introduced to enforce the cost constraint. This leads to the following primal-dual optimization formulation:

$$\min_{\lambda \geq 0} \max_\pi \left[ R(\pi) - \lambda(C(\pi) - c_{th}) \right] \tag{2}$$

which can be solved by iteratively updating the policy $\pi$ and the Lagrange multiplier $\lambda$. Specifically, $\lambda$ is updated by:

$$\lambda_{t+1} = \lambda_t + \alpha_\lambda(C(\pi_t) - c_{th}) \tag{3}$$

where $\alpha_\lambda$ is the learning rate.

**Online Safe RL.** Primal-dual based algorithms have shown great effectiveness and superior performance in the literature for online safe RL, which can combine a wide range of online unconstrained RL algorithms with the Lagrange multiplier method to create online safe RL algorithms. Without loss of generality, we consider SAC-lag (Ray et al., 2019) as the online algorithm, a primal-dual based algorithm that integrates the widely used SAC algorithm (Haarnoja et al., 2018) with the Lagrange multiplier method. More specifically, SAC minimizes the following objectives for the actor

(policy) and the critic (Q-function), respectively:

$$\mathcal{L}_\pi^{SAC}(\theta) = \mathbb{E}_{s\sim d}\mathbb{E}_{a\sim\pi_\theta(\cdot|s)}[\alpha\log\pi_\theta(a|s) - Q(s,a;\mu)] \tag{4}$$

$$\mathcal{L}_Q^{SAC}(\mu) = \mathbb{E}_{(s,a,s')\sim d}[(\hat{Q}(s,a;\mu) - y(r,s'))^2] \tag{5}$$

where $y(r,s') = r + \gamma\mathbb{E}_{a'\sim\pi_\theta(\cdot|s')}[\hat{Q}(s,a';\mu') - \alpha\log\pi(a'|s')]$, $Q(s,a;\mu)$ is parameterized by $\mu$, $\hat{Q}(s,a;\mu')$ is the target reward Q-function parameterized by $\mu'$, $d$ represents the data distribution in the replay buffer, and $\alpha > 0$ is some constant. To be applied in online safe RL, SAC-lag adapts SAC by using the Lagrangian method, resulting in the policy optimization objective as follows:

$$\mathcal{L}_\pi^{SAC}(\theta) = \mathbb{E}_{s\sim d}\mathbb{E}_{a\sim\pi_\theta(\cdot|s)}[\alpha\log\pi_\theta(a|s) - (Q(s,a) - \lambda Q_c(s,a))] \tag{6}$$

The optimization of the Q-functions for both reward and cost in SAC-lag is with Eq. (5) in SAC.

**Offline Safe RL.** Offline safe RL algorithms typically push up the cost estimations of OOD actions to avoid exploration beyond the offline dataset $\mathcal{D}$. Considering the comprehensive performance across various environments, in this paper we consider the SOTA Lagrangian-based algorithm for offline learning, namely CPQ (Xu et al., 2022). More specifically, CPQ first generates OOD actions via a conditional variational autoencoder (CVAE). Then, the cost of the generated OOD actions is increased by minimizing the following loss function for cost critic ($Q_c$-function):

$$\mathcal{L}_{Q_c}^{CPQ}(\mu_c) = \mathbb{E}_{(s,a,s')\sim d}\left[\left(Q_c(s,a;\mu_c) - \left(r + \gamma\mathbb{E}_{a'\sim\pi_\theta(\cdot|s')}[\hat{Q}_c(s,a';\mu_c')]\right)\right)^2\right] - \psi\mathbb{E}_{a\sim d,a\sim\nu}[Q_c(s,a;\mu_c)]$$

where $Q_c(s,a;\mu_c)$ is parameterized by $\mu_c$, $\hat{Q}_c(s,a;\mu_c')$ is the target cost Q-function parameterized by $\mu_c'$, $\nu$ represents the distribution of OOD actions generated by the CVAE. Additionally, to ensure both constraint safety and in-distribution safety, CPQ updates the reward critic (Q-function) using only state-action pairs that satisfy the cost threshold $l$:

$$\mathcal{L}_Q^{CPQ}(\mu) = \mathbb{E}_{(s,a,s')\sim d}\left[\left(Q(s,a;\mu) - \left(r + \gamma\mathbb{E}_{a'\sim\pi_\theta(\cdot|s')}[\mathbb{I}(Q_c(s',a';\mu_c) < l)Q(s,a';\mu)]\right)\right)^2\right]$$

where $\mathbb{I}(\cdot)$ is the indicator function, used to filter state-action pairs that satisfy the safety constraints. The policy loss function is given by:

$$\mathcal{L}_\pi^{CPQ}(\theta) = -\mathbb{E}_{s\sim d}\left[\mathbb{E}_{a\sim\pi_\theta(\cdot|s)}[\mathbb{I}(Q_c(s,a;\mu_c) < l)Q(s,a,\mu)]\right] \tag{7}$$

Similarly, when maximizing the reward, the policy only considers state-action pairs that meet the safety constraints. By assigning a higher cost to OOD actions, CPQ mitigates the OOD problem while meeting safety constraints.

**O2O Safe RL.** To the best of our knowledge, Guided Online Distillation (Li et al., 2024) is the only work studying O2O safe RL, which leverages a large-scale DT based on GPT-2 (Radford et al., 2019) as a guide policy to accelerate online learning, by following the idea of Jump-start RL (Uchendu et al., 2023). However, how to achieve fast safe online learning by finetuning a pre-trained policy is still not clear. Our work seeks to fill this gap and serves as an initial attempt to spur more interesting studies on policy-finetuning based O2O safe RL without using large models. A more detailed description of related work is delegated to Appendix B.

In this work, our objective is to enable faster and safer policy learning with standard online safe RL methods, by finetuning the policy and Q-functions pretrained using offline safe RL. In principle, any offline safe RL algorithms that output an offline policy and Q-functions can be used here for offline training.

## 3 WARM-START SAFE RL WITH VALUE PRE-ALIGNMENT

As demonstrated in Fig. 1, naively finetuning the offline policy for safe RL would not work well and the finetuned policy shows clear "inertia" in improving its performance: within a long period after online finetuning starts, its cost stays far below the limit, but its reward is quite low and not improving at all. This implies that such a strategy automatically "inherits" the conservatism from offline safe RL and is reluctant to actively explore in order to fully utilize the safe gap below the cost limit. In this section, we delve into the failure of naive finetuning, which points to two unique challenges for policy finetuning in O2O safe RL, i.e., erroneous offline Q-estimations and Lagrange multiplier mismatch. To address these problems, we propose a framework for O2O safe RL, namely warM-stArt safe Reinforcement learning with Value prE-aLignment (Marvel).

### 3.1 PRE-FINETUNE PHASE

*Challenge I: Erroneous Q-estimations resulted from objective mismatch and offline cost sparsity.* By learning from a fixed dataset without online environment interactions, offline safe RL typically suffers from large extrapolation errors for OOD actions beyond the support of the dataset. A general

principle to handle this is to penalize the reward/cost estimations for the OOD actions in such a way that risky explorations outside the dataset are discouraged. Particularly, the optimization of Q-functions in offline safe RL can be captured as follows:

$$\text{Offline } (Q): \min \mathbb{E}_{(s,a,r,s')\sim\mathcal{D}} \left[ \left( Q(s,a) - \left( r + \gamma \mathbb{E}\left[ \max_{a'} Q(s',a') \right] \right) \right)^2 \right] + \psi \cdot \mathcal{P}(s, a_{OOD}),$$

$$\text{Offline } (Q_c): \min \mathbb{E}_{(s,a,c,s')\sim\mathcal{D}} \left[ \left( Q_c(s,a) - \left( c + \gamma \mathbb{E}\left[ \max_{a'} Q_c(s',a') \right] \right) \right)^2 \right] - \psi_c \cdot \mathcal{P}_c(s, a_{OOD}).$$

Here $\psi \cdot \mathcal{P}(s, a_{OOD})$ and $\psi_c \cdot \mathcal{P}_c(s, a_{OOD})$ are the penalty terms. For instance, penalties are introduced in VOCE (Guan et al., 2024) so as to minimize the expected reward Q-values and maximize the expected cost Q-values for OOD actions. CPQ (Xu et al., 2022) increases the perceived cost of OOD actions during Q-function and policy updates, while keeping cost below the threshold.

In contrast, the optimization of Q-functions in online safe RL is standard without any penalty terms:

$$\text{Online } (Q): \min \mathbb{E}_{(s,a,r,s')\sim\mathcal{D}} \left[ \left( Q(s,a) - \left( r + \gamma \mathbb{E}\left[ \max_{a'} Q(s',a') \right] \right) \right)^2 \right],$$

$$\text{Online } (Q_c): \min \mathbb{E}_{(s,a,c,s')\sim\mathcal{D}} \left[ \left( Q_c(s,a) - \left( c + \gamma \mathbb{E}\left[ \max_{a'} Q_c(s',a') \right] \right) \right)^2 \right].$$

Obviously, Offline and online safe RL have distinct objectives for Q-functions, meaning pretrained Q-functions may not accurately estimate values for state-action pairs encountered during online interactions. As a result, offline policies tend to act overly conservatively, exploring only low-cost regions during online finetuning. However, effective online learning requires identifying state-action pairs with both high rewards and low costs, which necessitates exploring areas with potentially higher costs. This objective mismatch is even more pronounced in O2O safe RL due to the additional Q-function for cost estimation. The sparsity of offline cost leads to a pretrained cost Q-function that predicts low costs for IND state-actions, further limiting exploration during finetuning.

*Solution: Value Pre-Alignment.* To address the first challenge, a naive approach is to reevaluate the offline policy in online environments using Monte Carlo simulations, which however introduces additional interaction costs. Motivated by the recent advances in Off-Policy Evaluation (OPE) (Uehara et al., 2022), we borrow the idea from Fitted Q Evaluation (Hao et al., 2021) to align the offline Q-functions with the online learning objectives for the offline policy, by using the offline dataset before online policy finetuning. In particular, we seek to minimize the following objectives for the reward and cost Q-functions by starting from the pretrained Q-functions from offline learning, respectively:

$$\mathcal{L}_Q^{VPA}(\mu) = \mathbb{E}_D \left[ \mathcal{L}_2 \left( Q(s,a;\mu) - (r + \gamma \mathbb{E}_{a'\sim\pi_\theta(\cdot|s')}[\hat{Q}(s',a';\mu') - \alpha^{VPA}\log\pi_\theta(a'|s')]) \right) \right] \tag{8}$$

$$\mathcal{L}_{Q_c}^{VPA}(\mu_c) = \mathbb{E}_D \left[ \mathcal{L}_2 \left( Q_c(s,a;\mu_c) - (c + \gamma \mathbb{E}_{a'\sim\pi_\theta(\cdot|s')}[\hat{Q}_c(s',a';\mu_c') - \alpha_c^{VPA}\log\pi_\theta(a'|s')]) \right) \right], \tag{9}$$

where $D$ represents the distribution of $(s, a, s')$ in the offline dataset, and $\mathcal{L}_2$ denotes the MSE loss. $\mu$ denotes parameters of Q function and $\hat{Q}$ is the target Q function. Entropy terms are introduced in VPA to 1) optimistically estimate rewards to encourage greater exploration during the early stages of finetuning and 2) pessimistically estimate costs to ensure the agent remains cautious about the cost threshold during exploration. Specificity, the entropy terms can result in **both** higher rewards and costs for state-action pairs with high entropy, where the pretrained policy is 'uncertain'. We set the coefficient $\alpha$ for the Qc-network lower than $\alpha_c$ for the Q-network to encourage higher exploration during the agent's finetuning process. During the initial stage of online finetuning, the agent prioritizes exploring these high-reward areas, even at a high cost, as reward maximization dominates due to the small Lagrangian coefficient. This process helps refine the Q values through interactions. The offline policy remains unchanged during VPA to preserve the knowledge extracted from offline data. As a result, the agent can potentially explore high cost areas, without being overly conservative.

To characterize the performance of VPA in correcting the Q-estimations, we leverage Spearman's rank correlation coefficient, which measures the strength and direction of a monotonic relationship between two ranked variables. The reason is that the relative ranking of Q-values are more important than the absolute values for policy update. Specifically, given a dataset collected by rolling out the offline policy in the environment, we compare the ranking of learned reward/cost Q-values before and after VPA with that of estimated actual return by using Monte Carlo simulations. A large Spearman's rank correlation coefficient implies that the distribution of learned Q-values is more aligned

Table 1: The Spearman's rank correlation coefficients of the $Q$-value and $Q_c$-value are evaluated in the BallCircle and CarRun environments. "Random" refers to rollouts where the offline policy starts from a randomly initialized state-action pair (may not in the offline dataset), while "Dataset" refers to rollouts starting from a state-action pair randomly selected from the offline dataset.

| | VPA | BallCircle | | CarRun | |
|---|---|---|---|---|---|
| | | random | dataset | random | dataset |
| Q-value | before | -0.2387 | -0.3852 | -0.1143 | -0.5078 |
| | after | 0.5661 | 0.8278 | -0.0125 | 0.8314 |
| Qc-value | before | -0.2521 | 0.1725 | -0.2431 | -0.4327 |
| | after | 0.3579 | 0.8252 | 0.1254 | 0.4937 |

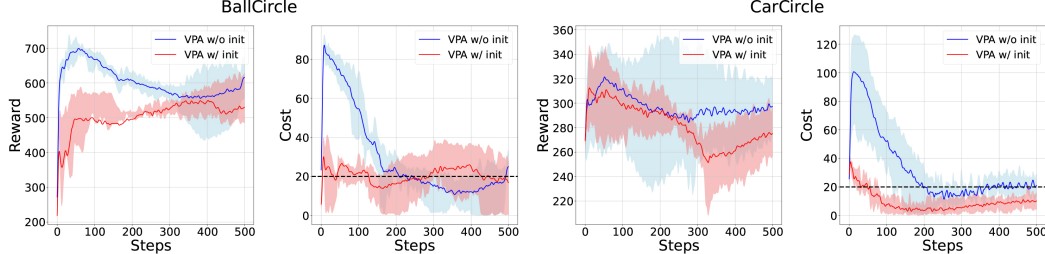

Figure 2: Comparison of online finetuning performance after VPA with two different initial values of the Lagrange multiplier. In 'VPA w/o init', the initial value is set to 0, whereas we initialize Lagrange multipliers with a good value found empirically (0.65 in BallCircle and 0.5 in CarRun) in 'VPA w/ init'. The multiplier is then updated using the standard dual ascent method.

with the distribution of true Q-values. As shown in Table 1, it is evident that the coefficient increases significantly after VPA for both reward and cost Q-values, no matter if the offline policy rolls out from a seen state-action pair in the offline dataset or from a randomly selected OOD state-action pair. This clearly demonstrates the effectiveness of VPA in aligning the pretrained Q-functions.

### 3.2 FINETUNE PHASE

*Challenge II: Lagrange multiplier mismatch.* Conventional value-based online safe RL relies on updating Lagrange multipliers alongside the policy and Q-functions during training, so as to push the overall cost below the limit while striking a right balance between maximizing the reward and minimizing constraint violations. While the policy and Q-functions can benefit from offline pretraining for a warm start, offline safe RL algorithms like CPQ (Xu et al., 2022) and BEAR-lag (Ray et al., 2019) cannot accurately estimate Lagrange multipliers with regularizing strengths matching with the cost of the offline policy, e.g., a small Lagrange multiplier is not power enough to push down the policy cost, while a large multiplier prevents active exploration of high-reward state-action pairs. For instance, in the BallCircle environment, the offline pretrained Lagrange multiplier value obtained using the BEAR-lag algorithm is approximately 1500, whereas during online finetuning, the SAC-lag requires a value of only about 0.65. The gap between these values clearly precludes the direct use of offline pretrained Lagrange multipliers. Improper initialization can lead to extensive constraint violations or training stagnation, an issue we term as the **Lagrange multiplier mismatch**.

On the other hand, as VPA promotes active exploration of high-reward state-actions by optimistically estimating rewards and pessimistically estimating costs, it inevitably increases the risk of exploring high cost state-actions, which in turn amplifies the need for appropriate Lagrange multipliers to quickly reduce the constraint violations. Figure 2 shows the online finetuning performance comparison after VPA in two environments between 1) empirically setting a good initial value for the Lagrange multiplier and 2) setting it to zero, where the traditional dual ascent method is used to update the multiplier. It is clear that a good initial multiplier can manage the cost very well, while the policy with a very small initial multiplier value suffers from large constraint violations and takes a much longer time to reduce the cost below the limit. The results also imply that although VPA aligns the distributions of Q-values, it may introduce high costs for online finetuning, which can be addressed with an appropriate initial Lagrange multiplier.

*Solution: Adaptive PID Control.* Clearly, finding a good initial value of the Lagrange multiplier can jointly address the mismatch problem and mitigate the potential risk of VPA. However, achieving this through experimental tuning is challenging, and currently, there is no theory to accurately predict

these values. To address this problem, instead of directly finding the best initial value, we take an alternative path by *quickly adapting* the Lagrange multipliers in an effective manner. Motivated by the recent success of leveraging PID control for updating the multiplier in online safe RL (Stooke et al., 2020; Yuan et al., 2022; Zhou et al., 2021), we introduce an adaptive PID control approach specifically tailored for O2O safe RL.

More specifically, compared to standard dual ascent in Eq. (3), PID control (Johnson & Moradi, 2005) offers a different approach to updating the Lagrange multiplier in the following way:

$$\lambda_{t+1} = \lambda_t + K_p e(t) + K_i \int_0^t e(\tau)d\tau + K_d \frac{de(t)}{dt} \tag{10}$$

where $e(t)$ captures the cost violation (errors) at time t, namely the cumulative cost difference from the policy rollout compared to the cost threshold. Here, $K_p$ is the proportional gain, corresponding to the instantaneous value of the error; $K_i$ is the integral gain to characterize the accumulation of past errors and $K_d$ is the differential gain corresponding to the rate of change of the error. In practice, the integral and differential are usually discretized as follows:

$$\int_0^t e(\tau)d\tau \approx \sum_{k=0}^t e(k)\Delta t, \quad \frac{de(t)}{dt} \approx \frac{e(t) - e(t-1)}{\Delta t}.$$

By taking the rate of change into consideration, the PID control can be especially useful in scenarios where costs fluctuate significantly, which however is not sufficient to handle the unique challenges in O2O safe RL. Specifically, to address the objective mismatch and the over conservatism exacerbated by sparse costs in the offline learning, VPA encourages active exploration but increases the risk of high cost. This requires a stronger control strength at the early stage of online finetuning to quickly reduce the cost. As training progresses and the cost is approaching to the limit, a weaker control strength is however preferred to stabilize the learning and keep the cost below the limit without large oscillations. The need of dynamic strength points to the need of adaptive control for O2O safe RL.

Towards this end, we propose an adaptive PID control approach for updating the Lagrange multiplier, which dynamically adjusts the PID control parameters during online finetuning based on the incurred policy cost over a time window of $n$ steps:

$$K_p \leftarrow \text{clip}\left(K_p \cdot \left(1 + \alpha \cdot \tanh\left(\frac{\bar{c}-c_{th}}{\bar{c}}\right)\right), K_{p_{min}}, K_{p_{max}}\right), \tag{11}$$

$$K_i \leftarrow \text{clip}\left(K_i \cdot \left(1 + \beta \cdot \frac{\bar{c}-c_{th}}{\bar{c}}\right), K_{i_{min}}, K_{i_{max}}\right), \ K_d \leftarrow \text{clip}\left(K_d \cdot \left(1 + \gamma \frac{\sigma_c}{\bar{c}}\right), K_{d_{min}}, K_{d_{max}}\right) \tag{12}$$

where the average cost $\bar{c} = \frac{1}{n}\sum_{i=1}^n c_i$, the standard deviation $\sigma_c = \sqrt{\frac{1}{n-1}\sum_{i=1}^n (c_i - \bar{c})^2}$, $\alpha$, $\beta$ and $\gamma$ are hyper-parameters. The design rationale is as follows: 1) By introducing the non-linear hyperbolic tangent function to adjust $K_p$, the Lagrange multiplier can respond quickly to large errors while avoiding frequent adjustments and reducing oscillations when the cost is close to the limit. When the average cost exceeds the limit ($\bar{c} > c_{th}$), $K_p$ and $K_i$ are increased to enhance the sensitivity of the PID controller to accelerate error correction. On the other hand, when the average cost is below the limit ($\bar{c} < c_{th}$), $K_p$ and $K_i$ are decreased to prevent overreaction of the PID controller, thus avoiding unnecessary oscillations and ensuring stability. By dynamically adjusting $K_p$ and $K_i$ in this manner, the controller adapts its low-frequency gain to match the current error magnitude, balancing speed and stability in error correction. 2) Because $K_d$ captures the volatility of cost changes, it will be adjusted using the standard deviation $\sigma_c$ of the cost over the period to stabilize cost control. A larger $\sigma_c$ indicates greater fluctuations in the error signal, suggesting the presence of high-frequency disturbances or noise. $K_d$ will be correspondingly increased, such that the controller can enhance its damping characteristics, adding phase lead and improving transient response. This helps mitigate the effects of sudden changes and stabilize the system.

In a nutshell, combining VPA and aPID leads to our proposed framework Marvel: 1) Given the pre-trained policy and Q-functions from offline learning, Marvel first applies VPA to align the pretrained Q-functions for both reward and cost using the offline dataset; 2) Marvel next utilizes Lagrangian-based online safe RL algorithms to further finetune both the pretrained policy and aligned Q-functions, by using aPID to update the Lagrange multipliers. We present the algorithmic framework of Marvel in Appendix A. Here VPA and aPID work in concert to guarantee the superior performance of Marvel: aPID addresses the Lagrange multiplier mismatch problem and quickly pushes down the potential high cost resulted by VPA, whereas VPA facilitates active exploration of high-reward state-action pairs and the usage of the pretrained policy as a warm-start for fast online finetuning with aPID control.

## 4 EXPERIMENTS

In this section, we conduct extensive experiments to verify the effectiveness of our approach, aiming to answer the following questions: 1) **RQ1:** How does our method compare with naive finetuning and other SOTA baselines in both reward and cost? 2) **RQ2:** How do different components in Marvel affect the performance? Due to the space limit, we delegate the experimental details and some additional results to Appendix D and Appendix F.

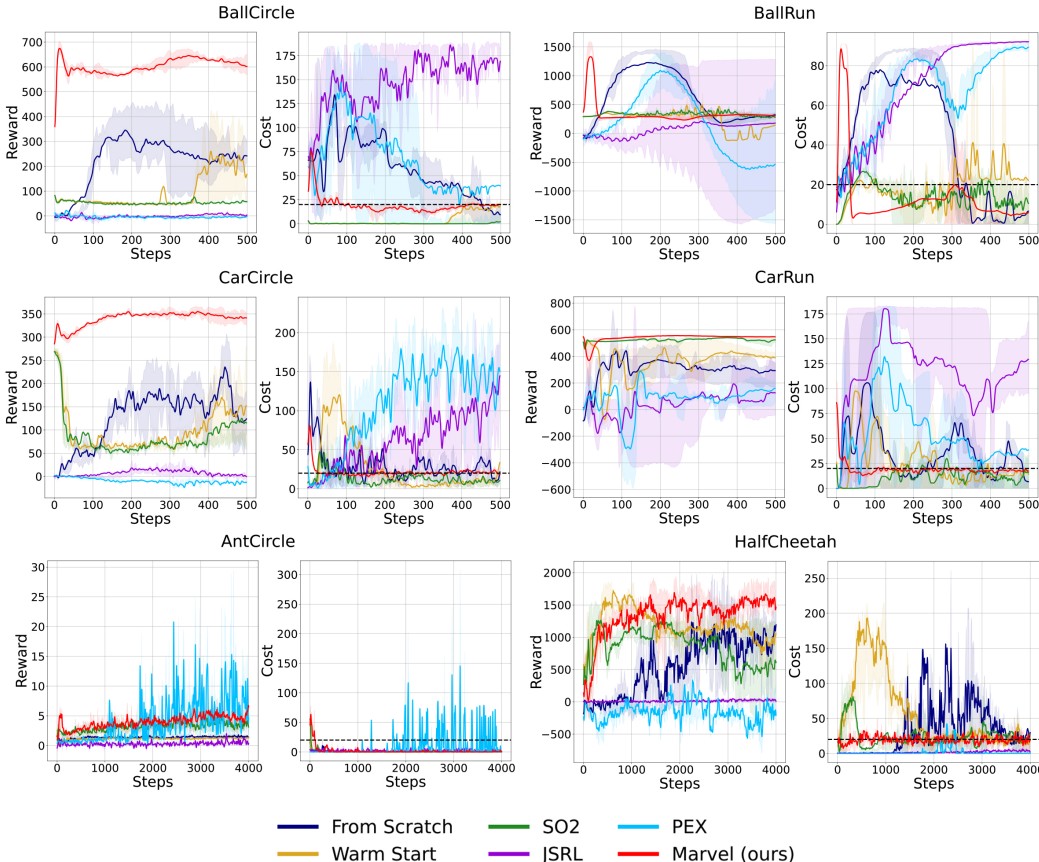

Figure 3: Performance comparison between Marvel and baseline methods in multiple environments. It is clear that Marvel can quickly find a high-return policy while keeping the cost below the limit.

### 4.1 EVALUATION SETUP

*Benchmarks.* We consider the DSRL benchmark (Liu et al., 2023b) and select **ten** environments from the Bullet Safety Gym (Gronauer, 2022) and Safety Gymnasium (Ji et al., 2023): BallRun, BallCircle, CarRun, CarCircle, HalfCheetah, AntCircle, AntRun, DroneCircle, Hopper, and Swimmer (results for the last four are in Appendix D.1). The cost threshold is set to be 20 in these environments. As mentioned earlier in Section 2, we choose CPQ and SAC-lag as base algorithms in our proposed framework Marvel for offline training and online finetuning, respectively, due to the effectiveness and representativeness of them. Each experiment was conducted using five random seeds, and the results were averaged to generate the final learning curves. We use a dataset that includes data provided by DSRL (Liu et al., 2024) and random data generated by a random policy to control the quality of the offline dataset.

*Baselines.* While Guided Online Distillation (Li et al., 2024) is the only work studying O2O safe RL, its usage of large pretrained model leads to an unfair comparison with standard RL frameworks using typically small-scale policy networks. In this work, we compare Marvel with **JSRL** (Uchendu et al., 2023), as Guided Online Distillation mainly follows this approach except using DT as the pretrained policy. Besides, we further adapt some SOTA approaches in O2O RL to O2O safe RL, including **SO2** (Zhang et al., 2024) and **PEX** (Zhang et al., 2023a), and a **Warm Start** approach as baselines. SO2 improves Q-value estimation through Perturbed Value Updates, JSRL and PEX utilize offline pretrained policies for exploration, and Warm Start directly finetunes the policy and

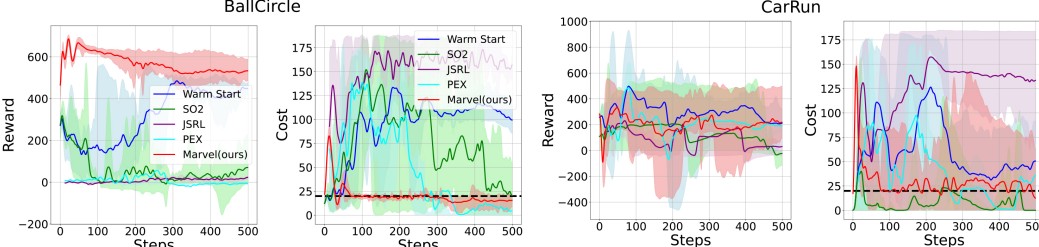

Figure 4: The policy pretrained with BEAR-lag performs poorly in cost, as this algorithm was not designed for safe RL. However, Marvel still achieves good results. This also indicates that Marvel performs well with pretrained policies of varying initial performance.

Q-networks from offline safe RL without modifications. We also compare with online learning from scratch, namely **From Scratch**. *More importantly, aPID is used to update Lagrange multipliers in all these baseline methods, which in fact already improves the performance of these methods compared to their original designs*. These baselines provide a meaningful comparison to demonstrate the effectiveness of Marvel in O2O safe RL. We provide the detailed descriptions of each baseline algorithm in Appendix C.

### 4.2 MAIN RESULTS

As shown in Fig. 3, Marvel demonstrates better or comparable performance compared to all baselines consistently across all environments, i.e., achieving the higher return while keeping the cost below the threshold. In stark contrast, the naive warm start method proves largely ineffective, often causing performance drop or stagnation during training. Without aligning the Q-estimations, both JSRL and PEX struggles a lot to improve during online learning and fails to control the cost. Besides, PEX also suffers from poor training stability and high variance across different settings. While SO2 mitigates the inaccuracies of Q-estimations related to O2O RL, it does so only to a limited extent and cannot maintain its performance consistently across different environments, although aPID has already been used to boost its performance. On the other hand, the fact that SO2 performs better than other baselines further indicates 1) the great potentials of enabling fast and safe online learning through policy finetuning (compared to using the pretrained policy only as a guide policy as in JSRL and PEX) and 2) the need of correcting pretrained Q-estimations before online finetuning.

More importantly, Marvel shows the superior capability of finding a good and safe online policy *very quickly* by using only a few steps of online interactions. In particularly, in environments like BallCircle and CarCircle, Marvel finds a good policy within less than 15 steps, dominating baseline methods in both performance and speed. *Note that all approaches indeed start from the offline policy and Q-functions, i.e., the same point at step 0 (ignored in all figures).* Guided by the offline policy and aligned Q-functions, Marvel rapidly jumps into the high-reward region in the state space, which highlights the effectiveness of VPA in addressing the overly conservative nature of offline pretrained policies. But this may also lead to a high cost at the beginning, e.g., the cost spike in the early step of finetuning as illustrated in Fig. 3. Because a few steps of online finetuning will just modify the policy in a neighborhood of the pretrained policy, aPID will lead the finetuned policy to low cost state-actions in the high-reward region by adaptively pushing the cost towards the threshold. It is also worth to note that we use the **same aPID parameters across all tested environments** without any further adjustments, demonstrating the robustness and effectiveness of our design.

**Compatibility of Marvel:** In O2O safe RL, compatibility with different offline safe RL methods is essential. Given the non-interactive nature of offline training and the potential unavailability of algorithms due to privacy concerns, this compatibility becomes even more critical compared to online algorithms. Our design of Marvel naturally fits a variety of offline safe RL methods and only requires a pretrained policy and Q-functions. To further verify this, Fig. 4 shows the training process using BEAR+Lagrangian (BEAR-lag) (Kumar et al., 2019) in the offline phase and SAC-lag in online finetuning. Note that BEAR-lag was not specifically designed for offline safe RL, but rather it incorporates the Lagrange multiplier into offline RL. In the BallCircle environment, Marvel achieves the highest reward while satisfying cost constraints, and in the CarRun environment, it also outperformed others while maintaining cost below the threshold. This highlights the flexibility of our algorithm across different offline safe RL methods.

### 4.3 ABLATION STUDIES

To answer **RQ2**, we conduct experiments in various setups. As shown in Fig. 5, the performance is best when both VPA and aPID are used. In contrast, if only VPA is used with traditional dual ascent during online finetuning, it significantly slows down safe online learning and takes a much longer

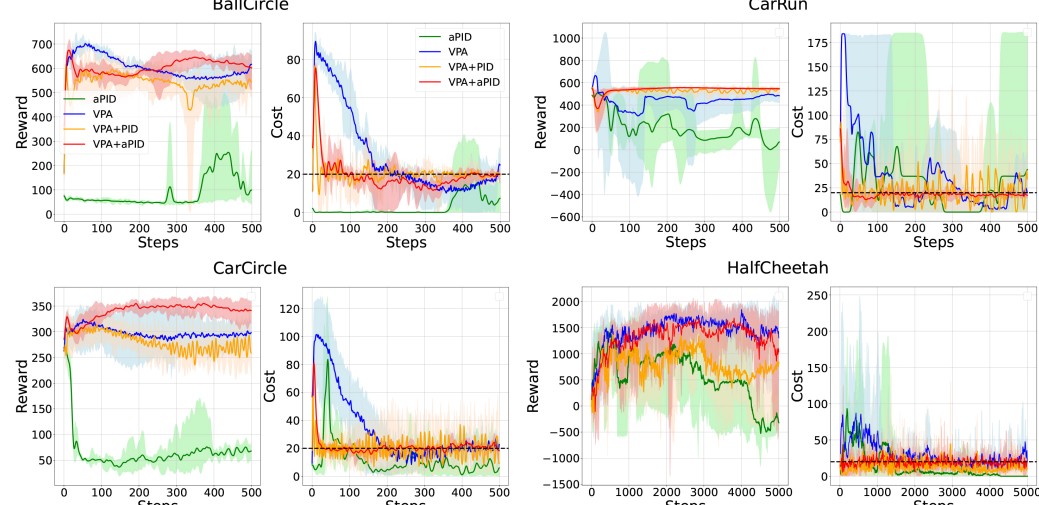

Figure 5: 'aPID' represents the scenario where aPID is used exclusively during the online finetuning phase, without applying VPA. 'VPA', on the other hand, shows the case where aPID is not used, and VPA is applied only during the pre-finetuning phase. VPA+PID use PID control for finetuning, while VPA+aPID employs aPID with adaptive parameter adjustment. Clearly, VPA+aPID achieves the best performance in terms of learning performance, speed, and stability.

time to reduce the cost. If only aPID is applied without VPA, the learning performance is very similar to naive policy finetuning, which struggles to improve due to the erroneous Q-estimations. We also evaluate the effectiveness of adaptive control in aPID, by comparing the performance between Marvel (VPA+aPID) and Marvel with aPID replaced by PID (VPA+PID). It can be seen from Fig. 5 that the training curve for cost exhibits significant fluctuations without using aPID. More critically, when the cost is close to the limit, PID cannot reduce its control strength. As a result, even if on average the cost of VPA+PID is close to the threshold, it is very frequent that the real-time cost exceeds the limit substantially, which is in fact not safe. Moreover, the inability to adjust the Lagrange multiplier promptly and appropriately affects the weight of reward and cost in policy updates, thereby influencing reward performance, as shown in the plots for CarCircle and HalfCheetah.

We also provide additional ablation studies in the Appendix D.2, which can provide the following insights: 1) Applying VPA to both the pretrained reward and cost Q-functions achieves the best performance compared to applying VPA to only one of them, which is reasonable as both will be manipulated during offline learning to reduce the extrapolation errors. 2) Marvel is robust to different qualities of the offline dataset. Regardless of the performance of the offline pretrained policy, Marvel can effectively finetune the policy while keeping the cost below the limit. 3) Introducing the entropy term in VPA is very helpful to improve the policy performance in terms of reward, by encouraging explorations of high-entropy states.

## 5 CONCLUSION

O2O safe RL has great potentials to put safe RL on the ground in real-world applications, by leveraging offline learning to facilitate fast online safe learning. In this paper, we proposed the first policy-finetuning based framework, namely Marvel, for O2O safe RL. In particular, by showing that naive finetuning would not work well, we identified two unique challenges in O2O safe RL, i.e., the erroneous Q-estimations and Lagrangian mismatch. To address these challenges, Marvel consisted of two key designs: 1) value pre-alignment to correct the Q-estimations before online finetuning, and 2) adaptive PID control to dynamically change the control parameters so as to rapidly and appropriately control the cost. Extensive experiments demonstrate the superiority of Marvel over multiple baselines. More importantly, Marvel is compatible to a variety of offline and online safe RL approaches, making it very practically appealing. For future work, it is interesting to take a closer look at the offline dataset, to identify states that are more worth exploring during online finetuning given the environmental information and cost threshold. Ultimately, we hope our work will bridge the gap between offline and online algorithms in safe RL, distinct from unconstrained RL, and enhance the efficiency of online safe RL, laying the foundation for the usage of safe RL in practical applications.

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

## A OVERVIEW OF ALGORITHM

---

**Algorithm 1:** Marvel

---

**Data:** Offline RL algorithm $\{L_{off}^{Q_\phi}, L_{off}^{Q_{c_{\phi_c}}}, L_{off}^{\pi_\theta}\}$, Online RL algorithm $\{L_{on}^{Q_\phi}, L_{on}^{Q_{c_{\phi_c}}}, L_{on}^{\pi_\theta}\}$

**Result:** Offline dataset $\mathcal{D}_{off}$, Online dataset $\mathcal{D}_{on}$, Network parameters $\phi, \phi_c, \theta$, Lagrange
        multiplier $\lambda$

**while** *in offline training phase* **do**
    $\phi \leftarrow \phi - \lambda_Q \nabla_\phi L_{off}^{Q_\phi}$;
    $\phi_c \leftarrow \phi_c - \lambda_Q^c \nabla_{\phi_c} L_{off}^{Q_{c_{\phi_c}}}$;
    $\theta \leftarrow \theta - \lambda_\pi \nabla_\theta L_{off}^{\pi_\theta}$;
**end**
% VPA with offline dataset;
**while** *in VPA phase* **do**
    **for** *each VPA step* **do**
        Sample transitions $(s, a, r, c, s') \sim \mathcal{D}_{off}$;
        Update $Q$ by Eq. (8), Update $Q_c$ by Eq. (9);
    **end**
**end**
**while** *in online training phase* **do**
    **for** *each environment step* **do**
        $a_t \sim \pi_\theta(s_t), s_{t+1} \sim T(s_t, a_t)$;
        $\mathcal{D}_{on} = \mathcal{D}_{on} \cup \{s_t, a_t, r(s_t, a_t), c(s_t, a_t), s_{t+1}\}$;
    **end**
    **for** *each update step* **do**
        $\phi \leftarrow \phi - \lambda_Q \nabla_\phi L_{on}^{Q_\phi}$;
        $\phi_c \leftarrow \phi_c - \lambda_Q^c \nabla_{\phi_c} L_{on}^{Q_{c_{\phi_c}}}$;
        $\theta \leftarrow \theta - \lambda_\pi \nabla_\theta L_{on}^{\pi_\theta}$;
        % update Lagrange multiplier with aPID;
        Update $\lambda$ by Eq. (10);
        update PID parameters by Eq. (11) and Eq. (12);
    **end**
**end**

---

## B RELATED WORK

**Online Safe RL.** Online safe RL approaches can be generally divided into several categories. The first category includes primal-dual based methods, such as PDO (Chow et al., 2018a), which combines PPO (Schulman et al., 2017) with the Lagrange multiplier method to obtain a policy that satisfies safety constraints. CPPO-PID (Stooke et al., 2020) combines PID control with Lagrangian methods to dampen cost oscillations. Similar Lagrangian-based methods are applied in conjunction with other unconstrained safe RL algorithms, such as TRPO-lag, PPO-lag, and SAC-lag. CPO (Achiam et al., 2017) inherits from TRPO (Schulman, 2015), optimizing with the Lagrange multiplier method within the trust region. CUP (Yang et al., 2022) extends CPO by incorporating the generalized advantage estimator. In comparison, RCPO (Tessler et al., 2018) uses different update rates for the primal and dual variables. Two-stage iterative methods have also been developed for online safe RL, e.g., PCPO (Yang et al., 2020) and FOCOPS (Zhang et al., 2020). Besides the primal-dual based methods, primal methods, which are also known as Lyapunov methods, have been leveraged in some studies for online safe RL. For instance, IPO (Liu et al., 2020) uses logarithmic barrier functions. P3O (Zhang et al., 2022) employs an exact penalty function to derive an equivalent unconstrained objective and restrict policy updates within the trust region. Chow et al. (2018b) leverages Lyapunov functions to handle constraints, which contains two parts, safe policy iteration and safe value iteration. Additionally, some studies (Wabersich et al., 2023; Choi et al., 2020) borrow techniques from the control theory, such as HJ reachability (Bansal et al., 2017; Yu et al., 2022) and control barrier functions (Ames et al., 2019), to ensure state-wise zero costs.

**Offline Safe RL.** Offline safe RL seeks to learn a safe policy from static datasets without online environmental interactions. Similar to online safe RL, Lagrangian methods can still be applied here, by adapting offline unconstrained RL algorithms like BCQ (Fujimoto et al., 2019) and BEAR (Kumar et al., 2019) to the safe RL setting. CPQ (Xu et al., 2022) uses a VAE to detect OOD (Ren et al., 2019) actions and penalizes them in terms of cost. COptiDICE (Lee et al., 2022a) extends OptiDICE (Lee et al., 2021) by adding safety constraints and derives a safe policy through the stationary distribution of the optimal policy. FISOR (Zheng et al., 2024) decouples the process of satisfying safety constraints from maximizing rewards and employs a diffusion model as the policy. VOCE (Guan et al., 2024) estimates Q-values of both cost and reward in a pessimistic way, mitigating extrapolation errors caused by OOD actions. Decision transformer (DT) (Chen et al., 2021) has also been applied to safe RL, leading to constrained decision transformer (Liu et al., 2023c).

**O2O Unconstrained RL.** O2O RL has recently attracted much attention in the unconstrained case, where a policy pretrained on an offline dataset is used to assist online policy learning, e.g., through finetuning or serving as a guide policy. More specifically, Hester et al. (2018); Nair et al. (2018); Rajeswaran et al. (2017) and Rudner et al. (2021) explore various combinations of offline demonstration data with online learning. The core idea is that pure offline RL often struggles with limited performance due to heavy reliance on dataset quality. However, if interaction with the environment is allowed, pffline pretrained policy can be finetuned for improved performance. However, naive implementation of this process often leads to suboptimal performance (Nair et al., 2020; Uchendu et al., 2023). AWAC (Nair et al., 2020) prioritizes actions with high advantage estimates, while AW-Opt (Lu et al., 2022) builds on AWAC by applying positive sample filtering and using hybrid actor-critic exploration during online finetuning. Lee et al. (2022b) finetunes the pretrained policy by balancing the offline and online datasets. FamO2O (Wang et al., 2024) trains a family of policies using a universal model and then employs a balance model to select the most suitable policy for each state. Cal-QL (Nakamoto et al., 2024) constrains the updates to the Q-network during online finetuning to prevent underestimation of the Q-values. SO2 (Zhang et al., 2024) improves Q-value estimation by updating Q-values more frequently and using noise-augmented actions. Instead of directly finetuning the pretrained policy, Jump-start RL (Uchendu et al., 2023) and PEX (Zhang et al., 2023a) follows another direction to leverage the offline policy, by using it to guide the update of the online policy during online learning.

## C  DETAILS ON BASELINES

Considering the characteristics of safe RL, which requires keeping the cost below a certain threshold, not all O2O unconstrained RL algorithms are suitable for O2O safe RL. For instance, AWAC (Nair et al., 2020), which maximizes the advantage function, has not yet been applied in the safe RL context. We compare Marvel with the following baselines:

**SO2** (Zhang et al., 2024). By analyzing Q-value estimation in offline to online transitions, the SO2 algorithm achieves more accurate Q-value estimation through Perturbed Value Update and by increasing the frequency of Q-value updates.

**JSRL** (Uchendu et al., 2023). JSRL employs an offline pretrained policy as the exploration policy and a policy under training during the online phase as the target policy. Initially, the exploration policy is used, followed by the target policy during online interaction to facilitate curriculum learning. To adapt to the safe RL setting, we update the Lagrange multipliers using the aPID method when updating the target policy.

**PEX** (Zhang et al., 2023a). Similar to JSRL, PEX uses an offline pretrained policy and a policy under training during the online phase for online interaction. However, PEX selects one of the actions based on the Q-networks's value estimation of actions chosen by the two policies. To meet the safe RL requirements concerning cost, like the modifications to JSRL, we use the aPID method to update the Lagrange multipliers.

**Warm Start**. We directly utilize the policy, Q-network, and Qc-network networks obtained from offline safe RL without any modifications (no VPA and aPID), and apply online safe RL algorithms for finetuning.

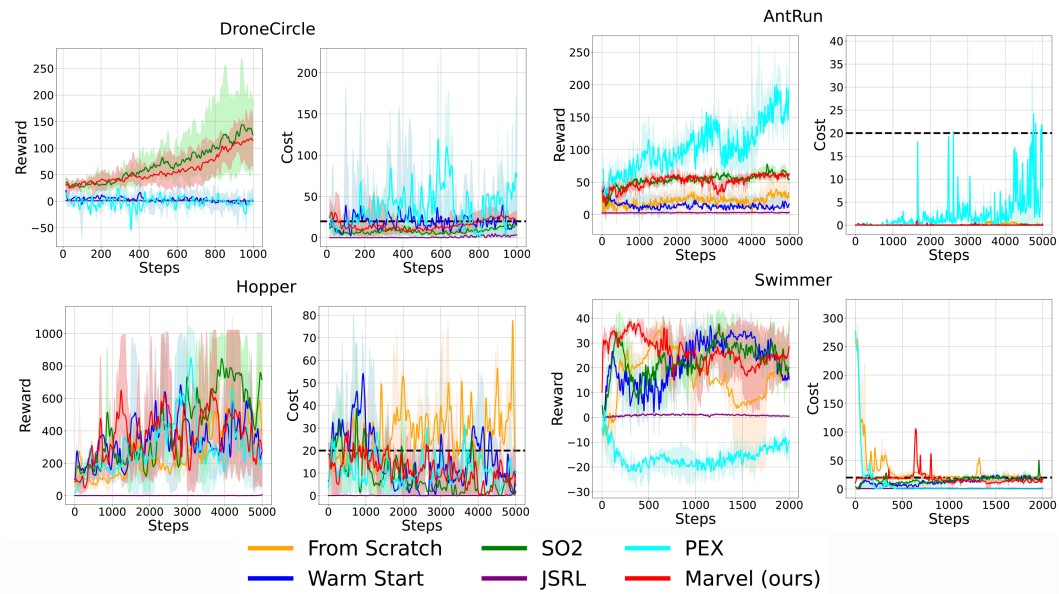

Figure 6: We provide experiments on more environments.

We selected SO2, JSRL, and PEX as baselines because they represent prominent methods in O2O RL, and adapting them to the safe RL context provides a meaningful comparison. Including these baselines allows us to demonstrate the effectiveness of Marvel in a fair and relevant context.

## D    MORE EXPERIMENTAL RESULTS

### D.1    MORE EXPERIMENTS

In Fig. 6, we additionally provide experimental results in more environments, including DroneCircle, AntRun, Hopper, and Swimmer. The results indicate that our proposed Marvel algorithm achieves competitive performance across these settings.

Additionally, we present the performance of the baseline algorithms and Marvel, along with the offline pretrained policy as the starting point, as shown in Fig. 3 and Fig. 6, and summarized in Table 2.

### D.2    MORE ABLATIONS

Fig. 7 presents more detailed ablation experiments, including whether VPA needs to be applied to both the Q-network and Qc-network, as well as whether the entropy term should be added to VPA. By comparing VPA(Q), VPA(Qc), and VPA(Q+Qc), we can observe that applying VPA solely to the Qc-network results in very poor performance during online finetuning. For example, in the Ball-Circle environment, results similar to naive finetuning shown in Fig. 1 were observed. On the other hand, applying VPA only to the Q-network leads to significant instability during finetuning (e.g., large error bands in BallCircle, CarCircle, and HalfCheetah) and poor performance in terms of cost (e.g., the cost curve in CarRun shows a sharp increase beyond the cost threshold). This occurs because if only the reward is optimistically estimated while the cost is pessimistically overestimated, it causes the agent to neglect the cost during exploration, adversely affecting finetuning performance. The experiments demonstrate that applying VPA to both the Q-network and Qc-network simultaneously has the best results, which aligns with the motivation discussed in Section 3. Comparing VPA(Q+Qc) with VPA(Q+Qc) with entropy, it is evident that optimistically estimating both reward and cost, while aligning with the pretrained policy, proves to be effective.

As shown in Fig. 8, regardless of the quality of the offline dataset or the performance of the pretrained policy, Marvel is able to quickly achieve optimal performance with only a few online interaction

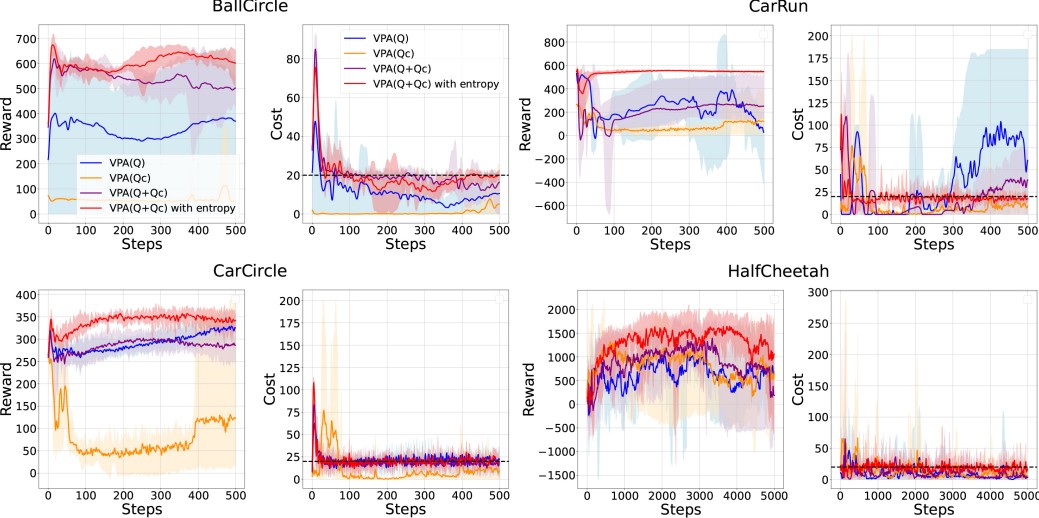

Figure 7: In the figure, VPA(Q), VPA(Qc), and VPA(Q+Qc) represent applying VPA to the Q-network, the Qc-network, and both simultaneously, without using the entropy term. This corresponds to setting $\alpha$ and $\alpha_c$ to 0 in Eq. (8) and Eq. (9). Conversely, VPA(Q+Qc) with entropy indicates that the entropy term is used in VPA, meaning $\alpha_c$ and $\alpha_c$ are non-zero. In all experiments represented by the curves, we employed aPID.

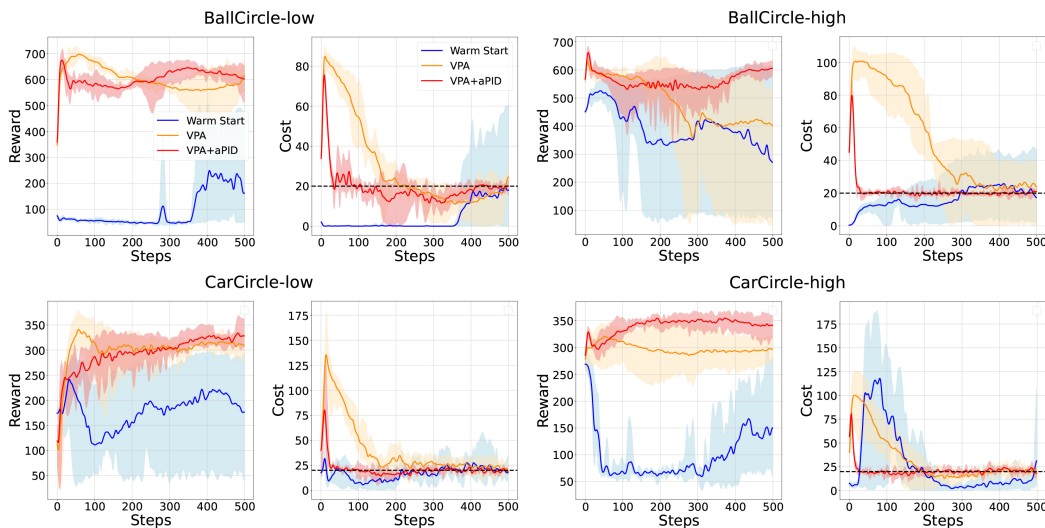

Figure 8: In the figure, "high" and "low" represent the different performance levels of offline pre-trained policies resulting from varying quality in the offline dataset. These policies are then finetuned online. The results demonstrate that the Marvel algorithm is robust to both different offline dataset qualities and pretrained policy performances.

steps. This highlights the robustness of the Marvel algorithm to variations in the quality of the offline dataset.

### D.3   FINETUNE Q-NETWORKS VS TRAIN NEW Q-NETWORKS IN VPA

In Marvel, VPA fine-tunes the offline pretrained Q-networks. Fig. 9 illustrates the training curves when, instead of fine-tuning the pretrained Q-networks, the Q-networks are retrained from scratch during the VPA phase and subsequently fine-tuned online. As shown, fine-tuning the pretrained Q-networks achieves better performance. This is because, although the pretrained Q functions may

be inaccurate, they still provide meaningful prior knowledge from the offline dataset and serve as a valuable starting point for Q function fine-tuning. This would generally speed up the learning and lead to a better local optima compared to learning from scratch based on the offline data from a random initial point. Moreover, considering the limited number of steps allowed in VPA for efficiency, directly learning completely forgoes the knowledge learned offline and can fail to find good Q estimations.

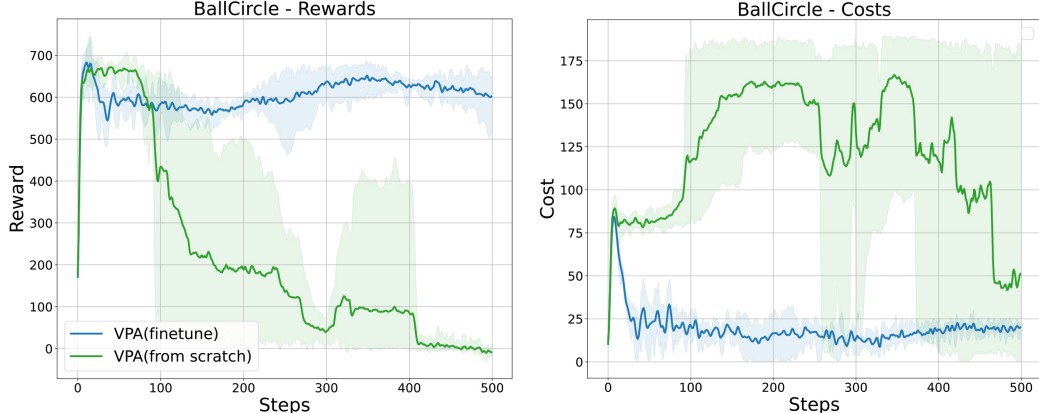

Figure 9: In the figure, "VPA (finetune)" refers to fine-tuning the offline pretrained Q-networks during the VPA phase, while "VPA (from scratch)" refers to training new Q-networks from scratch during the VPA phase.

### D.4   PARAMETER SENSITIVITY OF aPID

#### D.4.1   PID

The SAC-lag algorithm in Liu et al. (2024) utilizes PID control, with PID parameters carefully optimized. However, if their provided parameters are used directly under the environmental settings, policy updates, and Q-network update configurations of this paper, the performance is suboptimal. Fig. 10 presents a comparison, showing that when the PID parameters from the FSRL library are applied, the performance of online fine-tuning is significantly degraded. It is clear that the implementation of PID in our paper indeed significantly outperforms the implementation of PID provided by FSRL. More importantly, even with inappropriate PID parameters, aPID effectively boosts performance, achieving higher rewards while maintaining more stable cost levels.

#### D.4.2   PARAMETERS IN aPID

$\alpha$, $\beta$, and $\gamma$ are the parameters used in aPID to adjust the PID parameters. These parameters enhance the robustness of the initial settings for the PID parameters while being inherently robust themselves. Although our method aPID introduces more parameters, this is very common for adaptive algorithms in order to control the adaptation during the learning procedure. Fig. 11 illustrates the performance under various combinations of $\alpha$, $\beta$, and $\gamma$, with values ranging from 0.01 to 0.5. All curves achieve similar performance in terms of reward and cost by the end of training. This demonstrates that these parameters are both easy to tune and robust in their selection.

### D.5   PARAMETER SENSITIVITY OF VPA

The selection of $\alpha$ and $\alpha_c$ follows a similar approach to the selection of $\alpha$ in SAC. These values need to be empirically determined based on the evaluation results of the pretrained policy, the entropy of the policy, and the scale of the Q-values provided by the Q networks. It is crucial to ensure that the values of these parameters do not cause the entropy term to dominate the Q-value update process. The tuning process involves starting with small values, such as those in the range of $1 \times 10^{-5}$. Considering that the entropy value is typically a negative single-digit number, the upper limit for $\alpha$

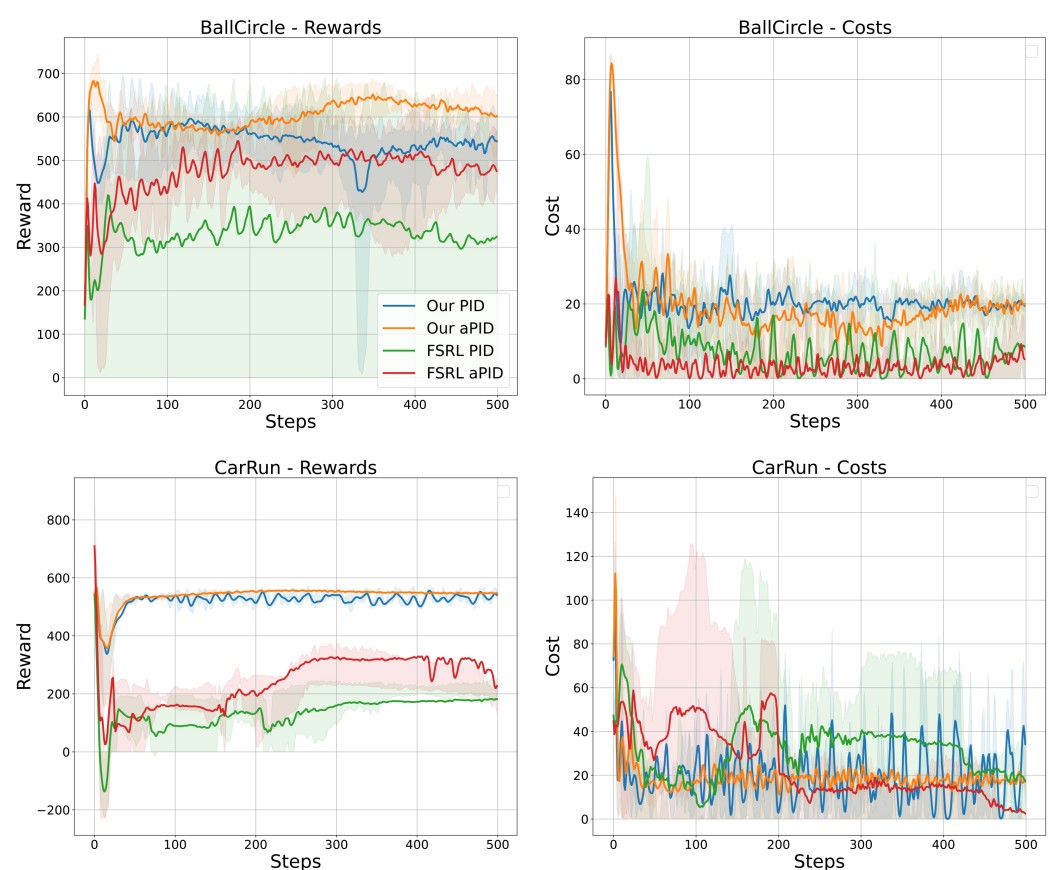

Figure 10: "Our PID" and "Our aPID" refer to using the PID and aPID parameters proposed in this paper for adjusting the Lagrange multipliers, respectively. Similarly, "FSRL PID" and "FSRL aPID" represent the parameters provided by the FSRL library for the same purpose.

and $\alpha_c$ should generally be around $1 \times 10^{-1}$. For relatively conservative offline pretrained policies, larger values of $\alpha$ and $\alpha_c$ may be more suitable.

To demonstrate the robustness of the chosen $\alpha$ and $\alpha_c$, we scaled the values provided in this paper by a factor of five, ranging from $1 \times 10^{-4}$ to $3 \times 10^{-5}$. As shown in Fig. 12, the choice of different $\alpha$ and $\alpha_c$ values has minimal impact on the final performance.

### D.6 CORRECTNESS OF OUR IMPLEMENTATION OF SAC-LAG

The primary goal of O2O safe RL algorithms is to achieve competitive performance with **minimal environment interactions** and in the **shortest time** by leveraging offline information to accelerate online learning. In contrast, the algorithm in (Liu et al., 2024) (and other similar online algorithms) achieves higher performance but relies on **significantly more interactions**. For example, in the BallCircle environment, (Liu et al., 2024) utilized 1.5 million environment interactions, whereas our method required only 120,000 interactions (with an average of 600 interactions per gradient update). This significant reduction highlights the efficiency of our approach, particularly in resource-constrained and safety-critical settings where the number of online interactions is strictly limited.

To validate the correctness of our implementation of SAC-lag, we conducted additional experiments comparing it to the SAC-lag implementation provided by the FSRL library under the same experimental settings (including both environment interaction steps and policy update frequencies). The results, presented in Fig. 13, show that both implementations demonstrate similar performance in terms of reward and cost. This validates the correctness of our implementation and ensures its reliability as a baseline for comparisons in our study.

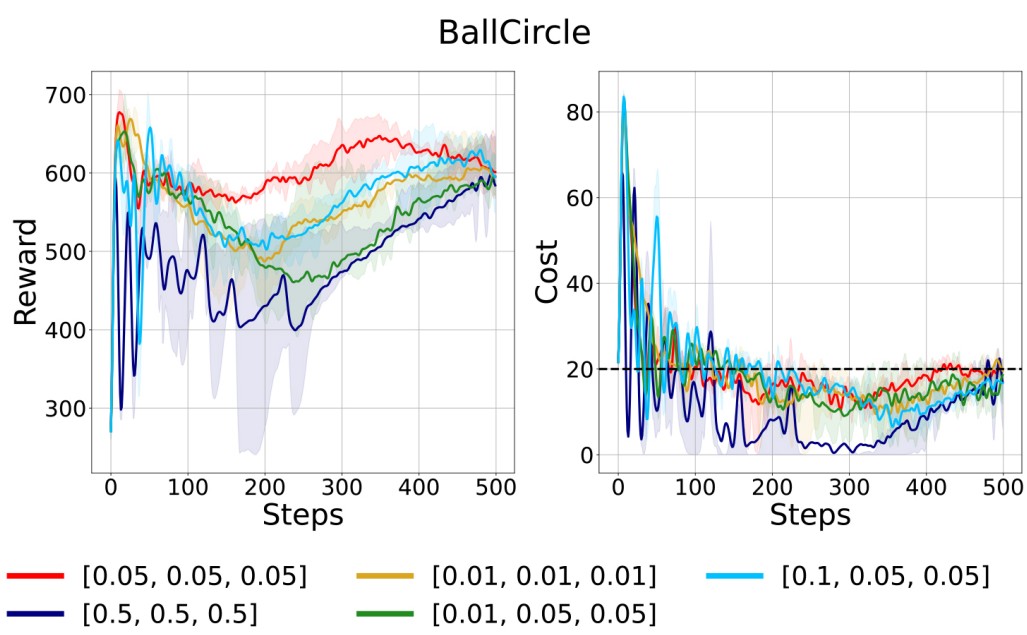

Figure 11: The three numbers within "[ ]" represent the three parameters for adaptively adjusting $K_p$, $K_i$ and $K_d$, namely $\alpha$, $\beta$ and $\gamma$.

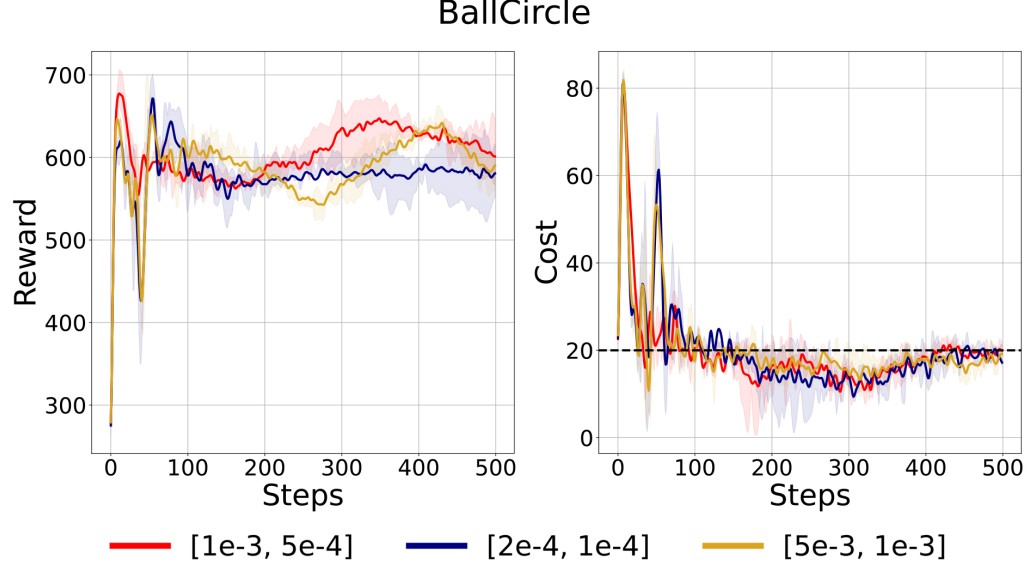

Figure 12: The two numbers inside "[ ]" represent the values of $\alpha$ and $\alpha_c$ used in VPA, as described in Eqs. (8) and (9).

## D.7 WITHOUT VPA BUT WITH GOOD INITIAL VALUES OF THE LAGRANGIAN MULTIPLIERS

As shown in Fig. 14, when VPA is not used and the Lagrange multipliers are updated using dual ascent (as described in Eq. (3)), even with appropriately chosen initial values for the Lagrange multipliers ("Warm Start w/ lag init"), the performance, while better than initializing with zero ("Warm Start"), still falls short of achieving optimal results.

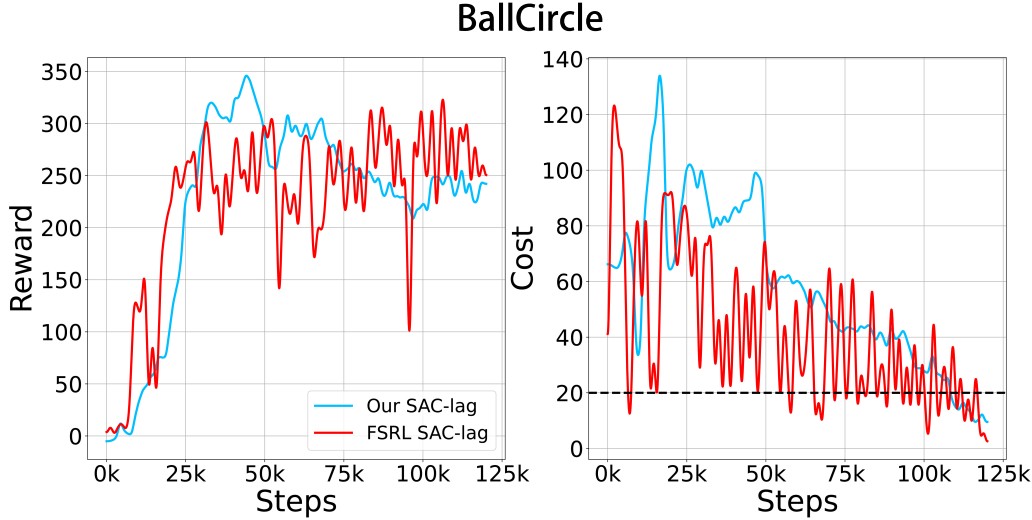

Figure 13: "Our SAC-lag" refers to the SAC-lag algorithm implemented in this paper, while "FSRL SAC-lag" represents the SAC-lag algorithm provided by the FSRL library. Using the same environment settings (including interaction steps) and update frequencies as in this paper, the results from the FSRL library are shown to be similar to ours.

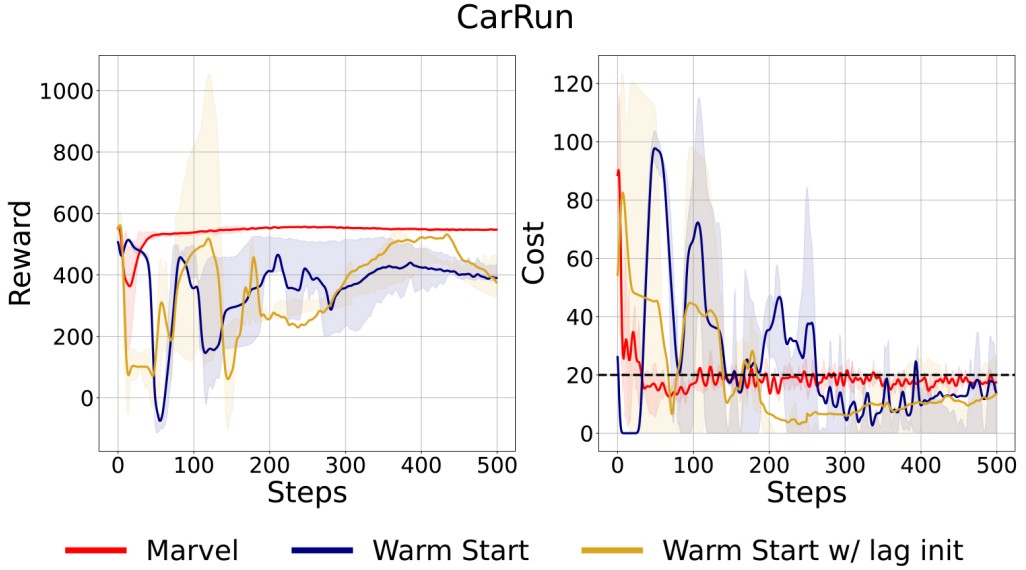

Figure 14: In the figure, "Marvel" and "Warm Start" follow the legend defined in Fig. 1. "Warm Start w/ lag init" represents the approach of empirically selecting appropriate initial values for the Lagrange multipliers and performing online fine-tuning.

## D.8 PERFORMANCE OF O2O RL WITHOUT CONSIDERING SAFE RL

In the O2O safe RL setting, when the O2O unconstrained RL algorithm is applied, the online fine-tuning results are shown in Figure Fig. 15. As can be seen, the unconstrained RL algorithm, which focuses solely on maximizing reward without controlling the cost, achieves a high reward but results in a cost that exceeds the threshold, thereby violating the constraint. This experiment further demonstrates the necessity of incorporating safe settings in such environments.

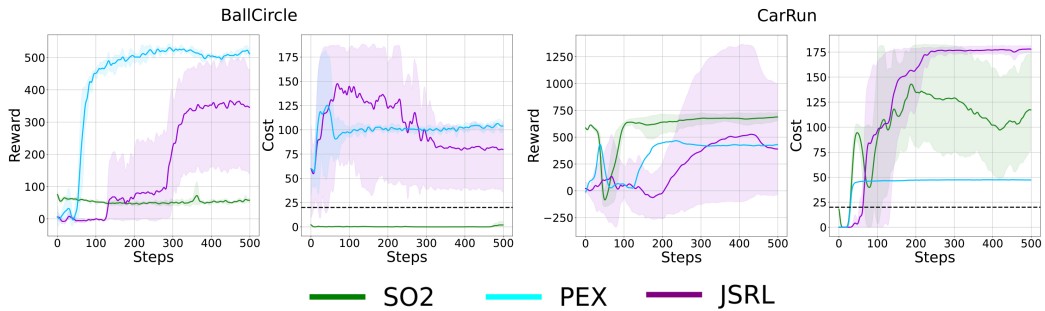

Figure 15: In the figure, the labels are consistent with those in Fig. 3.

# E    MORE ANALYSIS OF MARVEL

Similar to the analysis presented in Liu et al. (2023a), this section introduces an alternative way to evaluate safe RL performance beyond training curves, as shown in Fig. 16. The cumulative cost represents the total cost accumulated from all environment interactions up to a given timestep during training, while the max reward denotes the highest reward achieved up to that timestep. The relationship between these two metrics reflects the algorithm's ability to achieve maximum reward performance under a certain amount of cost incurred in the environment.

The figure shows that Marvel achieves the best max reward for a given cumulative cost. Moreover, when targeting a specific performance level (i.e., reward), Marvel requires the least cumulative cost. This further highlights Marvel's superior performance from another perspective.

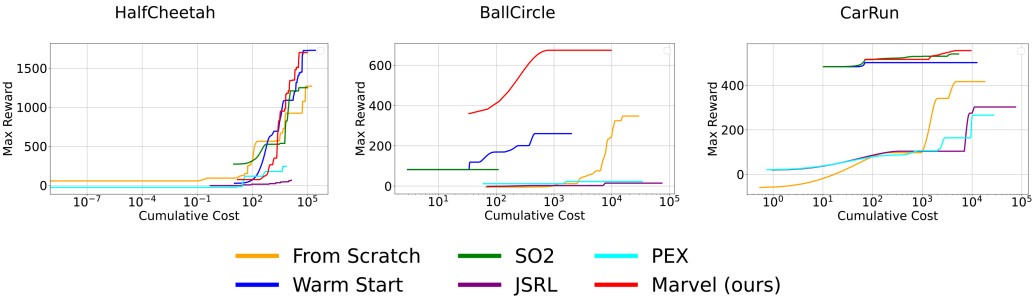

Figure 16: The legends in this figure follow the conventions of this paper and illustrate the relationship between cumulative cost and maximum reward.

# F    EXPERIMENTAL DETAILS

## F.1    SPEARMAN'S RANK CORRELATION COEFFICIENTS

We aim to explore the effect of VPA on the distribution of the Q and Qc networks. Specifically, for different state-action pairs, we need to analyze the true Q and Qc values versus the predicted values from the Q and Qc networks. To achieve this, we choose to use Spearman's rank correlation coefficient, which allows us to quantify the ranking accuracy of the Q and Qc values over a sequence of state-action pairs.

Spearman's rank correlation coefficient, denoted as $\rho$, is a non-parametric measure of the strength and direction of the association between two ranked variables. It evaluates how well the relationship between two variables can be described using a monotonic function, rather than assuming a linear relationship. This makes it particularly useful in our case, where we are more concerned with the rank ordering of predicted versus true Q and Qc values rather than their exact numerical differences.

Mathematically, Spearman's rank correlation coefficient is given by:

$$\rho = 1 - \frac{6 \sum d_i^2}{n(n^2 - 1)} \tag{13}$$

where $d_i$ is the difference between the ranks of the corresponding values of the two variables (in this case, the true and predicted Q or Qc values) for each state-action pair. $n$ is the number of state-action pairs.

Spearman's coefficient ranges from $-1$ to 1, where $\rho = 1$ indicates a perfect positive rank correlation (i.e., the predicted Q and Qc values perfectly match the rank of the true values) $\rho = -1$ indicates a perfect negative rank correlation, $\rho = 0$ indicates no correlation between the ranks of the predicted and true values.

By applying this measure, we can rigorously assess how well the Q and Qc networks preserve the relative rankings of the true values across various state-action pairs, thus quantifying the alignment between the predicted and true distributions.

### F.2 ESTIMATION OF Q-VALUES AND QC-VALUES THROUGH MONTE CARLO SIMULATIONS IN TABLE 1

The Q-values represent the expected cumulative reward from a given state when following a specific policy, while the Qc-values represent the expected cumulative cost. These values are estimated through Monte Carlo (MC) simulations, making them accurate because the simulations explicitly capture the sequential interactions of the agent with the environment under the given policy.

For the MC simulations, we use the pre-trained policy derived from the training phase. Each simulation starts from a selected initial state. The number of interaction steps with the environment depends on the specific settings of the environment. For instance, in the BallCircle environment, the maximum number of steps is 200. A total of 10 Monte Carlo simulations are performed, and at each timestep, we record both the reward and the cost. To compute the true Q-values and Qc-values, the recorded rewards and costs are averaged cumulatively across all steps in the episodes.

Regarding the choice of the initial state, the term "dataset" refers to selecting the initial state from the offline dataset used during VPA, whereas "random" indicates that the initial state is chosen randomly. This approach ensures a diverse evaluation and enhances the robustness of the estimated values.

### F.3 EXPERIMENTAL SETUP

In Table 3, we present the specific hyper-parameters used in the experiments. Table 4 lists the configurations of the environments used in the experiments.

## G LIMITATION

While Marvel performs well in most environments, it does not exhibit the same effectiveness in certain scenarios, such as in the AntRun environment. As depicted in Fig. 6, during finetuning, Marvel does not significantly improve cost and reward metrics. Consequently, aPID evidently does not function optimally in these settings. This suggests that further enhancements are needed for VPA to increase the agent's exploratory behavior during online finetuning. Combined with aPID's efficient control over costs, this approach could achieve optimal performance with minimal interaction with the environment, thus minimizing the time required.

Table 2: In the table, the content following "Offline" represents the performance of the pretrained policy, while the rest shows the results of online finetuning based on the pretrained policy using baseline methods and the Marvel algorithm.

(a) Result of Fig. 3

| Environment | Algorithm | Reward | Cost |
|---|---|---|---|
| BallCircle | Offline | 166.00 | 11.00 |
| | From Scratch | 241.70 | 10.94 |
| | Warm Start | 176.63 | 18.53 |
| | SO2 | 58.41 | 1.95 |
| | JSRL | 1.62 | 165.39 |
| | PEX | -4.27 | 39.85 |
| | Marvel (ours) | **603.94** | **19.75** |
| BallRun | Offline | 262.00 | 3.00 |
| | From Scratch | 315.21 | 5.58 |
| | Warm Start | 132.16 | 22.99 |
| | SO2 | 286.79 | 12.13 |
| | JSRL | 174.04 | 92.10 |
| | PEX | -555.58 | 88.84 |
| | Marvel (ours) | **306.55** | **5.48** |
| CarCircle | Offline | 265.00 | 14.00 |
| | From Scratch | 115.20 | 12.73 |
| | Warm Start | 141.55 | 24.05 |
| | SO2 | 115.91 | 9.07 |
| | JSRL | 1.06 | 130.01 |
| | PEX | -12.64 | 142.94 |
| | Marvel (ours) | **341.32** | **19.45** |
| CarRun | Offline | 544.00 | 72.00 |
| | From Scratch | 293.50 | 7.20 |
| | Warm Start | 391.40 | 16.90 |
| | SO2 | 522.23 | 16.22 |
| | JSRL | 126.59 | 128.21 |
| | PEX | 152.99 | 38.79 |
| | Marvel (ours) | **547.52** | **18.20** |
| AntCircle | Offline | 4.00 | 39.00 |
| | From Scratch | 1.48 | 0.01 |
| | Warm Start | 1.28 | 0.00 |
| | SO2 | 3.69 | 1.31 |
| | JSRL | 0.56 | 1.91 |
| | PEX | 6.93 | 2.27 |
| | Marvel (ours) | **5.56** | **0.87** |
| HalfCheetah | Offline | 113.00 | 17.00 |
| | From Scratch | 1074.27 | 28.81 |
| | Warm Start | 974.10 | 15.44 |
| | SO2 | 559.40 | 23.89 |
| | JSRL | 27.55 | 3.83 |
| | PEX | -155.56 | 0.18 |
| | Marvel (ours) | **1543.17** | **20.31** |

(b) Result of Fig. 6

| Environment | Algorithm | Reward | Cost |
|---|---|---|---|
| DroneCircle | Offline | 32.00 | 6.00 |
| | From Scratch | 0.77 | 28.81 |
| | Warm Start | 0.63 | 18.25 |
| | SO2 | 136.84 | 17.49 |
| | JSRL | 0.21 | 5.87 |
| | PEX | 3.22 | 74.86 |
| | Marvel (ours) | **117.54** | **22.19** |
| AntRun | Offline | 42.00 | 0.00 |
| | From Scratch | 30.87 | 0.11 |
| | Warm Start | 12.16 | 0.18 |
| | SO2 | 59.40 | 0.00 |
| | JSRL | 2.47 | 0.00 |
| | PEX | 175.63 | 18.27 |
| | Marvel (ours) | **61.05** | **0.00** |
| Hopper | Offline | 62.00 | 15.00 |
| | From Scratch | 467.07 | 35.19 |
| | Warm Start | 234.03 | 8.44 |
| | SO2 | 730.94 | 5.28 |
| | JSRL | 3.54 | 2.93 |
| | PEX | 215.63 | 12.32 |
| | Marvel (ours) | **310.82** | **7.09** |
| Swimmer | Offline | 4.00 | 2.00 |
| | From Scratch | 18.37 | 19.28 |
| | Warm Start | 17.04 | 18.80 |
| | SO2 | 18.73 | 19.59 |
| | JSRL | 0.11 | 0.00 |
| | PEX | -12.98 | 0.00 |
| | Marvel (ours) | **37.90** | **18.21** |

| Hyper-parameter | Value |
|---|---|
| Policy Learning Rate | 5e-5 |
| Q-network Learning Rate | 3e-5 |
| Qc-network Learning Rate | 8e-5 |
| Lagrangian Learning Rate | 1e-4 |
| SAC-lag: $\alpha$ | 5e-3 |
| VPA Entropy Coefficient : $\alpha$ | 1e-3 |
| VPA Entropy Coefficient : $\alpha_c$ | 5e-4 |
| aPID: Kp | 1e-4 |
| aPID: Ki | 1e-5 |
| aPID: Kd | 1e-5 |
| aPID: $\alpha$ | 0.05 |
| aPID: $\beta$ | 0.05 |
| aPID: $\gamma$ | 0.05 |
| Batch Size | 256 |
| MLP hidden layer size | [256, 256] |
| discount | 0.99 |
| $\tau$ | 5e-2 |
| replay buffer size | 1e6 |

Table 3: Experiment hyper-parameters

| Environment | Episode length | Cost threshold |
|---|---|---|
| BallCircle | 200 | 20 |
| BallRun | 100 | 20 |
| CarCircle | 200 | 20 |
| CarRun | 200 | 20 |
| AntCircle | 500 | 20 |
| AntRun | 200 | 20 |
| DroneCircle | 200 | 20 |
| HalfCheetah | 1000 | 20 |
| Hopper | 1000 | 20 |
| Swimmer | 1000 | 20 |

Table 4: Environment setup

