# OpenReview forum: "Marvel: Accelerating Safe Online Reinforcement Learning with Finetuned Offline Policy"
_ICLR.cc/2025/Conference — Submitted to ICLR 2025_

### Official Review · Reviewer_EtsL · 2024-10-26

**Soundness:** 2
**Presentation:** 3
**Contribution:** 2
**Rating:** 5
**Confidence:** 4

**Summary:**

This paper introduces the warM-stArt safe Reinforcement learning with Value prE-aLignment (Marvel) framework, featuring Value Pre-alignment (VPA) and Adaptive PID Control (aPID). VPA aligns pretrained Q-functions with true Q-values using offline data, promoting active exploration while managing costs. aPID adjusts Lagrange multipliers based on cost violations, ensuring stable online learning. Marvel outperforms baselines, achieving safe, high-reward policies with a few online interactions, and is compatible with many state-of-the-art safe RL methods.

**Strengths:**

1. The motivation for offline-to-online safe RL is clear, focusing on using offline data to accelerate online fine-tuning and reduce risky interactions in safety-critical scenarios.
2. The design of the method appears reasonable, introducing two key components that seem well thought out.
3. The results seem to indicate better performance, with evaluations conducted across various robots and tasks.

**Weaknesses:**

1. The method lacks theoretical justification regarding sample efficiency, leaving it unclear how it enables more efficient online finetuning.
2. The evaluation results are unconvincing, with certain key comparisons missing.

**Questions:**

1. For Value Pre-Alignment, how do the entropy terms in equations (8) and (9) contribute to achieving the desired optimistic/pessimistic estimation of rewards/costs? Besides, when calculating Spearman’s rank correlation coefficient, how are the "true" Q-values obtained or estimated?
2. For adaptive PID, could the authors explain why the non-linearity, i.e., tanh, is applied only to $K_p$ rather than to all $K_p$, $K_i$, and $K_d$, given that the authors state that the non-linearity can help respond quickly to larger errors while avoiding frequent adjustments and reducing oscillations (lines 351-354)? Additionally, the aPID introduces several more hyperparameters, e.g., the initial $K_p$, $K_i$, $K_d$, and their respective minimum and maximum values; how are these hyperparameters selected in practice, and is the proposed method sensitive to variations in their values?
3. For evaluation of O2O offline RL methods[1, 2, 3], typically it involves first assessing the performance of the offline pretrained policy (e.g., rewards and costs), followed by online finetuning, and then evaluating the finetuned policy to observe performance changes. To strengthen the analysis, could the authors provide: \
 a) Performance metrics for the offline pretrained policies prior to fine-tuning. \
 b) Details on the online finetuning process, including the number of online interaction steps and the cumulative cost, e.g., Figure 2 in [4]. \
 c) Comparison of the finetuned policy performance with pure online safe RL methods. \
 d) Analysis of whether the proposed method achieves higher rewards with fewer constraint violations. \
[1] https://proceedings.mlr.press/v202/yu23k/yu23k.pdf \
[2] https://arxiv.org/pdf/2110.06169 \
[3] https://arxiv.org/pdf/2201.10070 \
[4] https://proceedings.mlr.press/v202/liu23l/liu23l.pdf

---

> ### Author Response · Authors · 2024-11-20
> **response 1**
>
> Thank you for the insightful comments and constructive suggestions.
>
> **Q1: Unclear how it enables more efficient online finetuning**
>
> **A:** We have the following response to this point:
>
> 1) Offline-to-online RL has recently attracted much attention to facilitate more efficient online finetuning by leveraging offline data, as summarized in the related work in Appendix B. The general belief in this direction is that the offline dataset can provide much useful information and prior knowledge for online policy search, especially in the extreme case where an expert policy is learned from offline data. From a high level perspective, this is very similar to meta-learning where a good prior extracted from similar tasks can significantly speed up new task learning with only a few data samples. Beyond many experimental studies in this direction, multiple theoretical studies also confirm the great potentials of leveraging offline pretrained policies to facilitate efficient online learning, e.g., [1] and [2]. Our study follows a similar setup while seeking to address the unique challenges resulted in safe RL. More specifically, to first make the offline policy useful, accurate Q functions for this policy should be obtained, which is also consistent with the setups in [1] and [2]. Next, online policy update should be aware of increased risks of exploring high cost state-actions after we use VPA to correct offline Q estimations, and needs to response quickly and adaptively to the cost feedback during online update. From those perspective, we don't think that our method lacks theoretical justification and how it enables efficient online learning is unclear.
>
> [1] Bertsekas, D. Lessons from Alphazero for Optimal, Model Predictive, and Adaptive Control. Athena Scientific, 2022.
>
> [2] Wang, H., Lin, S., and Zhang, J. Warm-start actor-critic: From approximation error to sub-optimality gap. ICML, 2023.
>
> 2) Moreover, if the reviewer refers to theoretical evaluation of the sample complexity for online finetuning, we  thank the reviewer for this suggestion to further improve the quality of this paper. However, our main focus in this paper is to develop the first policy-finetuning based framework for offline-to-online safe RL by identifying and addressing the unique challenges therein, which is already nontrivial. As in many studies in offline-to-online RL, such as [1-3], no theoretical guarantees are provided along with the algorithm design. We think a comprehensive theoretical investigation deserves an independent study and will explore this in the future study. For this paper, we would appreciate it if the reviewer could evaluate our contributions from the problem formulation and algorithm design.
>
> [1] Zheng, H., Luo, X., Wei, P., Song, X., Li, D., and Jiang, J. Adaptive policy learning for offline-to-online reinforcement learning. AAAI, 2023.
>
> [2] Yu, Z., and Zhang, X. Actor-critic alignment for offline-to-online reinforcement learning. ICML, 2023.
>
> [3] Zhang, Y., Liu, J., Li, C., Niu, Y., Yang, Y., Liu, Y., and Ouyang, W. A perspective of Q-value estimation on offline-to-online reinforcement learning. AAAI, 2024.
>
> **Q2: Entropy terms in equations (8) and (9)**
>
> **A:** The entropy terms in equations (8) and (9) serve a similar role to the entropy terms in SAC. For state-action pairs where the policy provides high entropy, equations (8) and (9) will increase both the reward and cost. This encourages the agent to explore areas where the policy was "uncertain" after the offline phase, preventing the agent from becoming overly conservative during online fine-tuning and stagnating its performance near the offline-phase performance.
>
> **Q3: "true" Q-values in calculating Spearman’s rank correlation coefficient**
>
> **A:** These values are estimated through Monte Carlo (MC) simulations, making them accurate because the simulations explicitly capture the sequential interactions of the agent with the environment under the given policy.
>
> For the MC simulations, we use the pre-trained policy derived from the training phase. Each simulation starts from a selected initial state. The number of interaction steps with the environment depends on the specific settings of the environment. For instance, in the BallCircle environment, the maximum number of steps is 200. A total of 10 Monte Carlo simulations are performed, and at each timestep, we record both the reward and the cost. To compute the true Q-values and Qc-values, the recorded rewards and costs are averaged cumulatively across all steps in the episodes.

---

> ### Author Response · Authors · 2024-11-20
> **response 2**
>
> **Q4: Non-linearity in aPID**
>
> **A:** The proportional gain $K_p$ directly influences the response speed of the controller. The primary purpose of introducing the tanh activation function is to make the adjustment of the gain nonlinear within certain ranges, thereby allowing more precise control over the transition of the system’s response. This helps avoid excessive response or oscillation in certain cases. On the other hand, $K_i$ and $K_d$ are intended to reduce overall bias and excessive oscillation, rather than directly affecting the "intensity" of cost control. Therefore, there is less need to introduce non-linearity for these terms.
>
> **Q5: More hyperparameters in aPID**
>
> **A:** Although our method aPID introduces more parameters, this is very common for adaptive algorithms in order to control the adaptation during the learning procedure. Besides,  the additional parameters are very easy to tune and the performance of aPID is robust to the selection of these parameters. To show this, we have conducted additional results and added them to Appendix D.4.2 in the revision. In the experiments, we evaluate the performance of our method under a wide range of these parameters, from 0.01 to 0.5. It is clear that the performance of our method is consistently good under different parameter selections.
> Our additional results in Appendix D.4.1 demonstrate that even with suboptimal initial values for the PID parameters, using aPID  still leads to excellent performance.
> More importantly,  the same set of aPID parameters is used across all environments, which further justifies that the parameters are easy to tune.
>
> Thanks to the design of aPID, the parameter selection process is straightforward. Start by determining the parameters for the adaptive component with relatively small values, such as those suggested in this paper (0.05, 0.05, 0.05). Next, adjust the parameters for the PID component. Refer to parameters used in libraries like DSRL and FSRL, and follow traditional PID tuning rules, such as techniques from the Ziegler-Nichols method.
>
> Specifically, begin by setting the integral and derivative gains to zero, then gradually increase the proportional gain from zero. Once the optimal proportional gain is identified, proceed to adjust the integral and derivative gains, starting at approximately 0.05. After achieving the optimal settings, you can fine-tune the adaptive parameters based on the training curves for further refinement.
>
> **Q6: Evaluation of O2O offline RL**
>
> **A:** Thank the reviewer for the suggestions. We have revised our paper accordingly to include more analysis.
>
> a) All the training curves presented in the paper are from online fine-tuning. Therefore, the performance of the offline pretrained policies serves as the starting point for the training curves shown in the figures. For example, the performance of the pretrained policy for CPQ in the BallCircle environment is approximately a reward of 166 and a cost of about 11.
> Additionally, we provide the performance of the offline policy, which serves as the starting point for online finetuning, across all environments used in this paper in Appendix D.1.
>
> b) For all figures in the paper,
>  x-axis represents the number of gradient update steps of the policy. During each gradient update, the agent interacts with the environment for 3 episodes. For example, in the BallCircle environment, 3 episodes contain 600 environment interactions in total. The resulting states, actions, next states, rewards, and costs from the interactions are stored in the replay buffer. Then, the policy and Q-networks are updated. This entire process constitutes one gradient update.
> Besides, we  present a figure in Appendix E similar to Figure 2 in [4], which illustrates the relationship between cumulative cost and maximum reward. Notably, this figure reflects the cumulative cost required to achieve a certain level of performance as training progresses. It does not, however, show the reward and cost of the trajectories rolled out by an algorithm at a fixed number of training iterations, nor does it indicate whether the cost threshold has been violated.
>
>
> c) In Fig. 3 of the paper, we compare our methods with pure online safe RL, specifically SAC-lag.
>
> d) From the analysis in Figure 3 and Appendix E, similar to Figure 2 in [4], we can conclude that Marvel demonstrates strong performance in terms of both training steps and cumulative cost. Compared to baseline algorithms, Marvel achieves higher rewards while satisfying the cost threshold in a limited number of  training steps.  This demonstrates that **Marvel achieves higher rewards with fewer constraint violations**.

---

> ### Author Response · Authors · 2024-11-20
> **response 3**
>
> **end:**
>
> Finally, if our response resolves your concerns to a satisfactory level, we wonder if the reviewer could kindly consider raising the score of your evaluation. Certainly, we are more than happy to address any further questions that you may have during the discussion period. We thank the reviewer again for the helpful comments and suggestions for our work.

---

> ### Comment · Reviewer_EtsL · 2024-11-21
> **Additional clarifications are needed**
>
> I appreciate the authors' response and clarifications. However, my concerns and questions remain unresolved:
> 1) The proposed method appears to perform poorly after the offline pre-training phase. As shown in Table 2, the method either struggles to satisfy the cost threshold (e.g., CarRun and AntCircle) or fails to achieve high rewards, with a significant gap compared to the rewards reported in DSRL. For instance, in the DSRL paper, the rewards for safe trajectories satisfying the cost threshold of 20 are reported as 250+ for AntCircle (compared to 4 in Table 2), 2500+ for HalfCheetah (113 in Table 2), and 1500+ for Hopper (62 in Table 2). My question is: how can this poorly performing pre-trained policy "quickly find a safe policy with the best reward using only a few online interactions," as the authors claimed? In my opinion, while it is acceptable for an offline pre-trained policy to be sub-optimal, it should at least demonstrate sufficient performance to serve as a strong starting point for online fine-tuning.
> 2) I recommend that the authors include a table detailing the performance, such as rewards and costs, of both the offline pre-trained agents and the online fine-tuned agents. This would better illustrate the results, as the current figures are vague and do not allow for clear comparisons.
> 3) Regarding the entropy terms in equations (8) and (9), if the cost values of state-action pairs where the policy has high entropy increase, the policy will likely be updated to avoid those state-action pairs during learning. This is because the overestimated cost values would lead to constraint violations (i.e., $\mathbb{E}[ \text{cost} ] \leq \text{cost threshold}$). Consequently, this could result in an overly conservative policy. Could the authors clarify how this increased cost can 'encourage the agent to explore areas where the policy was uncertain after the offline phase,' and how it prevents the agent from becoming overly conservative during online fine-tuning?
> 4) Regarding the "true" Q-values in calculating Spearman’s rank correlation coefficient, the authors use the learned policy for MC simulation. However, typically, the "true" Q-values refer to the Q-values of the (unknown) behavior policy. Could the authors clarify why the Q-values of the learned policy can represent the Q-values of the behavior policy?

---

> > ### Author Response · Authors · 2024-11-22
> > **Response to 'Additional Clarifications Are Needed'**
> >
> > Thank the review for the new feedback.
> >
> > **Q1: Perform poorly after offline training**
> >
> > **A:** We have the following clarifications:
> >
> > 1) The offline pretraining part is **NOT** our design. Given offline pretrained policy and Q-functions, our method consists of two components, a) VPA which happens after offline training and before online finetuning, and b) aPID which is applied during online finetuning. In principle, any offline safe RL methods that output a policy and the corresponding Q functions can be used in the offline pretraining, which provide the starting points for our method.
> >
> > 2) It is clear that the performance of offline safe RL  heavily hinges upon the quality of the offline dataset and the learning algorithms. Without making the datasets, offline algorithms and  training setup consistent, directly comparing the numbers reported in DSRL and our experiments is not fair. 1) We use a mixed dataset, with random data generated through rollouts by a random policy, to regulate its quality.; 2) CPQ may not achieve the best performance
> >
> > 3) We kindly disagree with the reviewer that offline safe RL should demonstrate sufficient performance in order to serve as a starting point for online finetuning. In real applications, it is highly possible that offline data can be collected by some mediocre policy such that the pretrained policy can only have fair performance, which however doesn't mean that such a policy is useless or cannot be leveraged to facilitate faster online learning.
> > Even in the environments mentioned by the reviewer where our offline starting points are not good, our method still outperforms learning from scratch, which verifies the usefulness of these starting points.
> > More importantly, **it is NOT `the poorly performing pre-trained policy can quickly find a safe policy with the best reward using only a few online interactions'**; in contrast, **it is our method** that can quickly find a safe policy with the best reward **among all baselines** using only a few online interactions even starting from a poorly pretrained policy, based on the two key designs VPA and aPID to address the unique challenges in offline-to-online safe RL.
> >
> > 4) Besides, our design is independent from the quality of the offline pretrained policy, as we didn't make any assumption about the quality of the offline policy. That means, given a better offline policy as a starting point, our method will definitely work. For example, from the top-right plot of Fig. 8 (BallCircle-high), it can be observed that the pretrained policy achieves a reward of 450 and a cost of 0 in the BallCircle environment. (As mentioned in the paper, due to the drastic changes during the early stages of finetuning, the starting points of "Marvel" and "Warm Start" should be identical, but the visualization appears distorted.) This can be considered a good starting point for finetuning. From the training curves in the plot, it is evident that Marvel achieves excellent performance, reaching approximately 600 reward while satisfying the cost threshold.
> >
> > **Q2: Table**
> >
> > **A:** Thank the reviewer for this suggestion and also the suggestion of adding the cumulative cost in the first review to make our presentation more clear. We have added the table to Appendix D.1 (table 2).
> >
> > **Q3: Entropy**
> >
> > **A:** Thank the reviewer for the insightful comment. We would like to have the following clarification:
> >
> > 1) The entropy terms in eq. 8 and 9 will potentially lead to **both** higher reward and cost for state-action pairs where the pretrained policy is `uncertain', i.e., has high entropy values.
> >
> > 2) In fact, the high reward would play a more important role than the high cost, which can also be reflected by the fact that $\alpha$ in eq. 8 is larger than $\alpha_c$ in eq. 9 (as in Fig. 12 in the appendix). The agent would first explore these uncertain areas with high rewards during online finetuning, even if the associated costs can be high. This is  because the Lagrangian coefficient is too small at the initial stage of online finetuning and maximizing the Q values for the reward would dominate the policy optimization. This exploration will further correct the Q values through online interactions.
> > As a result, the agent can potentially explore high cost areas, without being overly conservative. We have also clarified this in our revision.
> >
> > **Q4: MC simulation**
> >
> > **A:** To clarify this, it is important to understand the purpose of VPA, because calculating the correlation is used to evaluate the effectiveness of VPA. Here VPA is proposed to correct the pretrained Q values for the pretrained policy, to make these Q values closer to **the true values of the pretrained policy** (not some unknown behavior policy as mentioned by the reviewer). That is why we use the pretrained policy for MC simulations, which could give us an estimate of the true Q values for the pretrained policy. We have never claimed that we aimed to estimate the true Q values for some behavior policy.

---

> ### Author Response · Authors · 2024-11-25
>
> Dear Reviewer,
>
> As the author-reviewer discussion period will end soon, we will appreciate it if you could check our response to your review comments. This way, if you have further questions and comments, we can still reply before the author-reviewer discussion period ends. If our response resolves your concerns, we kindly ask you to consider raising the rating of our work. Thank you very much for your time and efforts!

---

### Official Review · Reviewer_KNkj · 2024-10-31

**Soundness:** 3
**Presentation:** 2
**Contribution:** 2
**Rating:** 5
**Confidence:** 3

**Summary:**

This article systematically investigates the challenges of O2O (Offline-to-Online) in safe RL and proposes corresponding solutions. Specifically, the paper 1) employs value pre-alignment to counteract erroneous Q-estimations and 2) utilize adaptive PID control to overcome Lagrangian mismatch. The method introduced has demonstrated outstanding performance across datasets in DSRL.

**Strengths:**

1. The O2O in safe RL is a very important but overlooked issue. This paper systematically explores this issue.
2. The paper's writing logic is very clear, progressing methodically from problem analysis, to problem formulation, and finally to the solution. So the understanding is relatively easy.

**Weaknesses:**

1. This paper studies the problem of Offline to Online finetuning for safe RL. The first challenge can be viewed as a common issue in the O2O (Offline-to-Online) domain. The second challenge is unique to safe RL, but I question its significance. Figure 2 suggests that a good initialization of Lagrange multipliers is more conducive to ensuring a reasonable cost. However, even if the initial values of the Lagrange multipliers are not reasonable, they can be adjusted through Equation (3). The proposed aPID seems to merely expedite this adjustment process, as shown in Figure 5. In addition, all baselines involve aPID, but the performance is still poor. This also indicates that **challenge 2 is not crucial**.

2. Guided Online Distillation is an important baseline of this paper. Regarding the reason "its usage of large pretrained model leads to an unfair comparison with standard RL frameworks", I have some disagreements. Guided Online Distillation is based on decision transformer (DT). Despite having more parameters, DT is far away from a large model. Moreover, DT's performance on D4RL is typically weaker than that of standard RL algorithms, such as IQL.

**Questions:**

1. The meanings of 'Warm Start', 'From Stractch', and 'PID' are not clear in introduction, which makes the Figure 1 is hard to understand.

---

> ### Author Response · Authors · 2024-11-20
> **response**
>
> Thank you for the insightful comments and constructive suggestions.
>
> **Q1: Challenge 2 is not crucial**
>
> **A:** We kindly disagree with the reviewer about the two challenges and will have the following clarifications:
>
> Firstly, as we states clearly in the introduction, Challenge 1 arises not only because the objectives of offline and online safe RL are different, but also due to the sparsity of cost in the offline dataset. This is very different from the unconstrained case and will lead to extremely low cost for in-distribution state-actions. As a result, direct online finetuning will stay with extremely conservative low-cost policy and fail to leverage the room below the cost threshold to improve the reward (as shown in Fig. 1(a) about the performance of "Warm start").
>
> Second, in terms of Challenge 2, it is worth to note that the results in Fig. 2 are after we apply VPA to solve the first challenge. Without solving the first challenge using VPA, simply selecting good initial values of the Lagrangian multipliers or updating them based on Eq. 3 is not sufficient to guarantee the performance, especially considering that we need fast online adaptation with only a limited number of interactions in offline-to-online safe RL.
> We demonstrate this point through additional experiments shown in Appendix D.7.
> This points out the importance and necessity of using VPA to address challenge 1.
>
> Third, a side effect of using VPA is that it promotes active exploration of high-reward state-actions and inevitably increases the risk of exploring
> high cost state-actions, which  amplifies the need for appropriate Lagrange multipliers to
> quickly reduce the constraint violations for fast online learning. Solving the challenge 2 is important to guarantee the success of VPA. As  shown in Fig. 5, applying VPA without addressing challenge 2 cannot guarantee good performance.
> Moreover, we don't think that the fact that baseline methods still perform poorly after using aPID actually indicates that challenge 2 is not crucial. Instead, it indicates that solving challenge 2 alone is not sufficient and challenge 1 is also more severe in offline-to-online safe RL due to the sparse costs, so that we need to handle both challenges simultaneously.
>
> **Q2: Guided Online Distillation as an important baseline**
>
> **A:** We kindly disagree with the reviewer about this point. DTs in the original paper for Guided Online Distillation use large models such as pre-trained
> GPT2 as the network backbone. In stark contrast, the majority of algorithms in standard RL use small size neural networks. Marvel utilizes a simple neural network architecture with two hidden layers, each with size of 256.
> Compared to these small networks, the DT based on GPT2 is clearly a large-scale model. If we ignore this large-scale model and just replace it with a small neural network in Guided Online Distillation, the resulted approach is just JSRL, which was used as a baseline in our paper.
>
> **Q3: Meanings of 'Warm Start', 'From Scratch', and 'PID' are not clear**
>
> **A:** While these terminologies are well-known in the community of offline-to-online RL and control, we appreciate the reviewer a lot for pointing them out. The explanations are as follows and we will also modify the introduction to make them more clear:
> ‘Warm Start’ refers to using the offline pretrained policy and Q-networks to directly initialize an online safe RL algorithm. ‘From Scratch’ refers to training a purely online safe RL algorithm with randomly initialized policy and Q networks. ‘PID’ refers to a proportional–integral–derivative controller, which is used to adjust the values of Lagrange multipliers in primal-dual based algorithms.
>
> **end:**
>
> Finally, if our response resolves your concerns to a satisfactory level, we wonder if the reviewer could kindly consider raising the score of your evaluation. Certainly, we are more than happy to address any further questions that you may have during the discussion period. We thank the reviewer again for the helpful comments and suggestions for our work.

---

> > ### Comment · Reviewer_KNkj · 2024-11-30
> >
> > My primary concern was whether this paper merely represents a combination of safe RL and offline to online finetuning. However, the authors' response to **Q1** has largely alleviated my concerns. Consequently, I have decided to raise my rating to 5.

---

> > > ### Author Response · Authors · 2024-12-02
> > >
> > > Thank you for your response. We would appreciate the reviewer if the reviewer can point out any remaining concerns that we haven't solved.

---

> ### Author Response · Authors · 2024-11-25
>
> Dear Reviewer,
>
> As the author-reviewer discussion period will end soon, we will appreciate it if you could check our response to your review comments. This way, if you have further questions and comments, we can still reply before the author-reviewer discussion period ends. If our response resolves your concerns, we kindly ask you to consider raising the rating of our work. Thank you very much for your time and efforts!

---

### Official Review · Reviewer_DDV9 · 2024-11-01

**Soundness:** 3
**Presentation:** 3
**Contribution:** 3
**Rating:** 6
**Confidence:** 3

**Summary:**

This paper focuses on offline-to-online (O2O) safe reinforcement learning (RL). It identifies two main issues causing inefficiency in previous methods: erroneous Q-estimations and Lagrangian mismatch. To address these issues, the authors propose two methods: Value Pre-Alignment and Adaptive PID Control. These methods aim to improve the standard O2O safe RL framework. Several experiments provide evidence of the effectiveness of these two components.

**Strengths:**

1. The explanatory experiments in the Method section really enhance the reader's understanding of the proposed methods. Improvements in the clarity of some figures could further aid in the visualization of the results.

2. The paper is well-written, with each part of the methods meticulously described and well-motivated, contributing to a coherent and comprehensive presentation of the research.

**Weaknesses:**

1. The algorithms and baselines lack stability in some environments. More repeated experiments may be needed. The improvement over baselines is not significant, showing performance gaps only in two tasks: BallCircle and CarCircle.

2. All figures need to be polished, especially Figure 3. The legend should not be limited to one or two figures but could be placed below all figures. Additionally, the lines representing each method should be improved, as it is difficult to distinguish between methods in some figures.

**Questions:**

1. How would the method perform if equipped with a large-scaled model, such as the structure of Guided Online Distillation?

2. How does aPID improve baselines, as the paper claims it can enhance their performance?

3. Could other types of O2O baselines be considered, such as Online Decision Transformers [1], which do not involve Q-functions?

4. What are the base algorithms for Warm Start?

5. A guide for hyperparameter settings would be beneficial, such as for $\alpha$ and $\alpha_c$. Is the method sensitive to these hyperparameters?

[1] Zheng Q, Zhang A, Grover A. Online decision transformer[C]//international conference on machine learning. PMLR, 2022: 27042-27059.

---

> ### Author Response · Authors · 2024-11-20
> **response 1**
>
> Thank you for the insightful comments and constructive suggestions.
>
> **Q1: Lack stability in some environments**
>
> **A:** We would like to have the following clarifications. As shown in Fig. 3, 1) Marvel demonstrates superior stability compared to other baseline algorithms.
> Note that a key aspect of safe RL is ensuring safety, which requires keeping the cost below the predefined threshold while maximizing reward.
> Marvel achieves this balance effectively, as its cost remains below the threshold in all environments discussed in the paper.
> 2) In the meanwhile, Marvel consistently achieves
> the best reward among all baselines in the BallCircle, BallRun, CarCircle, CarRun, HalfCheetah, DroneCircle, and Swimmer environments. By maintaining safety and achieving exceptional rewards, Marvel demonstrates state-of-the-art performance in the majority of environments, providing strong evidence of its outstanding capability in offline-to-online safe RL.
>
> **Q2: Figures need to be polished**
>
> **A:** We appreciate the reviewer for this suggestion. We have further polished the figures in the revision to make them more clear.
>
> **Q3: Marvel in large-scaled model**
>
> **A:** In principle, Marvel can leverage any offline safe RL algorithms as the base component for offline learning, as long as a pretrained policy and Q functions are provided by the offline algorithm. Given the superior generalization capability of large-scale models, we expect that the performance of Marvel can be further boosted when equipped with large model based policy and Q functions. However, current large-scale models in reinforcement learning, such as Decision Transformer (DT), do not rely on explicit Q-network updates to adjust the policy. As a result, they are incompatible with the Marvel algorithm, which requires pre-adjustments to Q-networks before fine-tuning.
>
> **Q4: How aPID improves baselines**
>
> **A:** Offline-to-online safe RL has been rarely
> explored in the literature, so there are no other baseline approaches we can compare. To address this, we adapt state-of-the-art methods for offline-to-online RL in the unconstrained case based on the  Lagrangian method. However, their performance can be poor as these approaches were not originally designed for safe RL. Considering that aPID  can also be treated as a general enhancement of the dual ascent method in primal-dual approaches, we incorporate aPID into these baselines to further strengthen their performance.
> Specifically, for online policy updates in these baselines, we employed the Lagrange multiplier method, using aPID to dynamically update the Lagrange multipliers. However, despite the use of aPID, these adapted baseline algorithms still failed to achieve satisfactory performance and were outperformed by Marvel.
> This indeed indicates that simply using aPID in existing offline-to-online RL methods to solve the Lagrangian mismatch problem is not sufficient to guarantee good performance in offline-to-online safe RL, and both challenges identified in our paper need to be handled simultaneously.
>
> **Q5: Other baselines that do not involve Q-function**
>
> **A:** The Online Decision Transformer can be considered an O2O RL algorithm, but it is not applicable to safe RL settings. While the Constrained Decision Transformer is a DT variant designed for safe RL, it operates purely as an offline RL algorithm and cannot be applied in online scenarios. In other words, there are currently no suitable algorithms for O2O safe RL that do not rely on Q-networks. Therefore, it cannot be used as a baseline for comparison.
>
> **Q6: Base algorithms for Warm Start**
>
> **A:** The base algorithms for Warm Start (i.e., online finetuning) is SAC-lag.
>
> **Q7: Selection of $\alpha$ and $\alpha_c$ and if it’s sensitive**
>
> **A:** The selection of $\alpha$ and $\alpha_c$ is similar to the selection of $\alpha$ in SAC. These values need to be determined empirically based on the evaluation results of the pretrained policy, the entropy of the policy, and the scale of the Q-values provided by the Q networks. It is important to ensure that the values of both parameters do not cause the entropy term to dominate the Q-value update process. The tuning process involves starting with small values, such as those in the range of $1\times 10^{-5}$. Considering that the entropy value is typically a negative single-digit number, the upper limit for $\alpha$ and $\alpha_c$ should generally be around 0.1. For relatively conservative offline pretrained policies, larger values of $\alpha$ and $\alpha_c$ may be more suitable.
>
> To demonstrate the robustness of the chosen $\alpha$ and $\alpha_c$, we scaled the values provided in this paper by a factor of five, ranging from $1\times 10^{-4}$ to $3\times 10^{-5}$. As shown in Appendix D.5, the choice of different $\alpha$ and $\alpha_c$ values has minimal impact on the final performance.

---

> > ### Comment · Reviewer_DDV9 · 2024-11-26
> > **Response**
> >
> > I would like to thank the authors for the rebuttal that addressed most of my concerns. Please find my subsequent comments and a few additional questions below:
> >
> > I am curious about how O2O RL algorithms, such as the Online Decision Transformer, perform in these environments, even if they are not equipped with safe RL settings. I think that including such experiments could further demonstrate the necessity of safe settings in these environments.
> >
> > Additionally, I have a question for discussion purposes (not requiring further experiments) regarding large-scale models. The QT[1] method, which is based on the Transformer and incorporates the Q value, could serve as a baseline algorithm to evaluate the effectiveness of Marvel on large-scale models.
> >
> > [1] Hu S, Fan Z, Huang C, et al. Q-value regularized transformer for offline reinforcement learning[J]. arXiv preprint arXiv:2405.17098, 2024.

---

> > > ### Author Response · Authors · 2024-11-27
> > > **Response to additional questions**
> > >
> > > Thank the reviewer for the additional feedback.
> > >
> > > **Q1: Performance of O2O RL without considering safe RL.**
> > >
> > > **A:** Many thanks for this insightful comments. We have conducted additional experiments to directly evaluate the performance of O2O RL algorithms, such as JSRL, PEX and SO2, and put the results in Appendix D.8. As expected, while these methods can achieve high rewards, they suffer from high costs and cannot satisfy the safety constraints, which clearly indicates that these methods cannot be directly used in safe RL.
> > >
> > > **Q2: The QT method.**
> > >
> > > **A:** Many thanks for this suggestion.
> > > QT[1] enhances traditional Conditional Sequence Modeling-based DT methods by using Q-learning to learn an additional Q function, which improves the performance of DT during inference. It can be potentially used in our framework for offline pretraining, where VPA will work in concert to correct offline Q estimations and speed up better online policy learning. However, while aPID is specifically designed for primal-dual methods aimed at addressing safe RL, but QT, being a transformer-based method focused on unconstrained RL, may not leverage this design effectively.
> > > Given the large training cost this could incur, we will explore this baseline in our future revision to demonstrate the effectiveness of Marvel when equipped with large-scale models.
> > >
> > > Finally, we thank the reviewer again for this additional discussion to improve the quality of our paper, and we are more than happy to answer any further questions you may have.

---

> ### Author Response · Authors · 2024-11-20
> **response 2**
>
> **end:**
>
> Finally, if our response resolves your concerns to a satisfactory level, we wonder if the reviewer could kindly consider raising the score of your evaluation. Certainly, we are more than happy to address any further questions that you may have during the discussion period. We thank the reviewer again for the helpful comments and suggestions for our work.

---

> ### Author Response · Authors · 2024-11-25
>
> Dear Reviewer,
>
> As the author-reviewer discussion period will end soon, we will appreciate it if you could check our response to your review comments. This way, if you have further questions and comments, we can still reply before the author-reviewer discussion period ends. If our response resolves your concerns, we kindly ask you to consider raising the rating of our work. Thank you very much for your time and efforts!

---

### Official Review · Reviewer_ZJP2 · 2024-11-02

**Soundness:** 2
**Presentation:** 2
**Contribution:** 2
**Rating:** 5
**Confidence:** 4

**Summary:**

This paper addresses the challenges of safe RL by proposing Marvel, a framework that bridges offline and online RL to achieve safer and faster policy learning. Based on the authors’ claim, Marvel tackles these issues with two key components: Value Pre-Alignment, which adjusts Q-functions for accurate value estimation, and Adaptive PID Control, which optimizes Lagrange multipliers for balancing rewards and safety constraints during online fine-tuning. Experimental results show that Marvel outperforms presented baselines.

**Strengths:**

(1) Interesting topics: it focuses on an interesting topic: safety in reinforcement learning, specifically exploring offline-to-online adaptation to enable safer and more efficient policy learning—a crucial yet underexplored area in RL.

**Weaknesses:**

(1) Typo: In Lines 413 and 414, the HalfCheetah, Hopper, and Swimmer in DSRL are from Safety-gymnasium [1], not Bullet-Safety-Gym.

(2) Confusion about experiment results: Why do the results of training-from-scratch look so poor? Based on the report [1], SAC-Lag should achieve 600+ rewards on Ball-Circle and 350+ rewards on Car-Circle.

(3) Concerns about the experiment results. The experiment results shown in Figure 6 indicate that the proposed method can not reach a reasonable reward. For example, in Drone-Circle, the agent should achieve a reward of 600+ using SAC-Lag. However, the proposed Marvel only achieves 150-.

Reference:
[1] https://fsrl.readthedocs.io/en/latest/tutorials/benchmark.html

**Questions:**

(1) Can you clarify what the x-axis “steps” means in Figure 1 and Figure 2? Does it mean the optimizer update times?

(2) Can you compare your method to pure offline safe RL baselines such as CDT [1] and pure online safe RL baselines such as PPO-Lag and SAC-Lag?

Reference:

[1] Zuxin Liu, et al. "Constrained decision transformer for offline safe reinforcement learning." International Conference on Machine Learning. PMLR, 2023.

---

> ### Author Response · Authors · 2024-11-20
> **response**
>
> Thank you for the insightful comments and constructive suggestions.
>
> **Q1: Typo**
>
> **A:** Thank you for pointing this out. We have fixed the typo in the revision.
>
> **Q2: Experiment results**
>
> **A:** In terms of the performance of purely online learning,  we think the reviewer may misunderstand the advantage/setup of offline-to-online RL and hope our following explanations can clarify this.
>
> 1) The primary goal of offline-to-online algorithms is to achieve competitive performance with **minimal** environment interactions and in the **shortest time**, by leveraging offline information to speed up online learning. In contrast, the algorithm in [1] (almost the same for all the online algorithms) achieves higher performance but relies on **significantly more interactions**. For instance, in the BallCircle environment, [1] utilized 1.5 million interactions, whereas our method required only 120,000 interactions (with an average of 600 interactions per gradient update). This explains the performance difference relative to [1], and underscores the efficiency of our approach in resource-constrained and safety-critical settings where only a limited number of online interactions is allowed.
>
> 2) We have also done additional experiments to compare our implementation of SAC-lag with that provided by the FSRL library, under the environment and training setting in our paper (including both environment interaction steps and policy update frequencies). This result is shown in Appendix D.6 in the revision. It can be seen that both implementations indeed demonstrate very similar performance in terms of both reward and cost, which is sufficient to verify the correctness of our implementation of SAC-lag.
>
> **Q3: x-axis “steps” means in Figure 1**
>
> **A:** Sorry for the confusion. The "steps" on the x-axis represents the number of gradient updates of the policy (i.e., optimizer update times).
>
> During each gradient update, the agent interacts with the environment for 3 episodes. For example, in the BallCircle environment, 3 episodes contain 600 environment interactions in total. The resulting states, actions, next states, rewards, and costs from the interactions are stored in the replay buffer. Then, the policy and Q-networks are updated. This entire process constitutes one gradient update. We have clarified this in the revision.
>
> **Q4: Compare to pure offline and online safe rl**
>
> **A:** We have the following response:
>
> 1) Our method provides an offline-to-online framework for safe RL, which seeks to facilitate fast online learning by leveraging offline pretraining on a fixed dataset. The objective is different from pure offline safe RL, which seeks to learn a safe policy from offline data only, and also different from pure online safe RL, which does not leverage offline data at all and typically requires a lot of online interactions to find a good policy.
> Compared to pure offline safe RL, our method can improve the performance through online interactions and mitigate the impact of low-quality offline data. Compared to pure online safe RL, our method can quickly prompt a good policy with a few online interactions, which is very important for practical applications of safe RL.
> 2) Since our method only requires a pretrained policy and Q functions from offline learning, in principle any offline safe RL method that outputs both policy and Q functions can be leveraged in our method as a base component. We have compared our method with pure online safe RL baselines, e.g., SAC-Lag, and it is clear that our method can quickly find a good policy in terms of both reward and cost.
>
> **end:**
>
> Finally, if our response resolves your concerns to a satisfactory level, we wonder if the reviewer could kindly consider raising the score of your evaluation. Certainly, we are more than happy to address any further questions that you may have during the discussion period. We thank the reviewer again for the helpful comments and suggestions for our work.

---

> ### Author Response · Authors · 2024-11-25
>
> Dear Reviewer,
>
> As the author-reviewer discussion period will end soon, we will appreciate it if you could check our response to your review comments. This way, if you have further questions and comments, we can still reply before the author-reviewer discussion period ends. If our response resolves your concerns, we kindly ask you to consider raising the rating of our work. Thank you very much for your time and efforts!

---

### Official Review · Reviewer_tMnm · 2024-11-02

**Soundness:** 2
**Presentation:** 2
**Contribution:** 2
**Rating:** 5
**Confidence:** 4

**Summary:**

This paper aims to solve two issues in offline-to-online safe RL, Q value estimation error and Lagragian multiplier mismatch, and proposes a new algorithm. The proposed method first re-trains Q function with an online objective to correct Q estimation and then uses an adaptive PID control to update Lagrangian multiplier. The authors run experiment on several safe RL tasks and claim their method outperforms other baselines.

**Strengths:**

- Offline-to-online method is important to the practical application of safe RL.
- The two challenges pointed out by this paper (i.e., Q value estimation error and Lagragian multiplier mismatch) are insightful.
- The experiment is extensive, and the proposed method outperforms the compared baseline.

**Weaknesses:**

- Some concerns on VPA:
  - If you believe the pretrained Q functions from offline learning is not accurate, why not directly learn a new Q function by online objective or eq.(8)&(9) instead of training based on the pretrained Q functions? I believe it is a very straightforward baseline which should be compared.
  - The error in Q value estimation is not only from the mismatch between offline and online learning objective, but also from the distribution shift issue due to the online policy update. Can the proposed method mitigate error from this aspect?

- Although the experiments show that adapative PID Lagrangian updater can stablize the cost performance, its effectiveness is still questionable. The adaptive PID introduces three more hyperparameters $\alpha, \beta, \gamma$. Along with original $K_p, K_i, K_d$, there are **six** hyperparameters for the updater in total. However, the adaptive PID only shows a marginal improvement over PID in fig.5. Meanwhile, the original hyperparameters of PID are not well tuned. The authors select $K_p=10^{-4}, K_i=10^{-5}, ...$, which are not only very different from the original PID Lagrangian paper [1], but also different from the recent safe RL benchmarks using PID update [2][3], which provide more stable performances of PID update than the reported one in fig.5.

- The experiment results show that the proposed method outperforms other offline-to-online baselines or naively starting from offline policy. However, in the experiment, we can find those offline-to-online methods are even worse than purely online learning (i.e., learning from scratch). Therefore, I believe those baselines are not strong enough to illustrate the effectiveness of new method. Meanwhile, the performance of purely online learning is also significantly under-reported: SAC-Lag can quickly achieve ~700 reward with even smaller cost in Ballcircle according to [3]. It can also achieves > 2500 reward in Halfcheetach-velocity in [2].

minor issues:
- I don't think PPO ("Schulman et al 2017", the first citation in line 93) is a safe RL algorithm, it is a standard RL algorithm.

[1] Responsive Safety in RL by PID Lagrangian Methods

[2] OmniSafe: An Infrastructure for Accelerating Safe Reinforcement Learning Research, https://www.omnisafe.ai/en/latest/

[3] FSRL, https://fsrl.readthedocs.io/en/latest/index.html

**Questions:**

- What is the input to tSNE in fig.1(b)? Current caption is quite confusing, and I don't understand why the visualization shows cost is sparser.
- What are the $\mu$ and $\hat{Q}$ in eq.(8)?
- How do you get the true Q-values in table 1? E.g., what is the policy for MC simulation? how many simulations do you get for the Q computing? And are the state-action pairs for evaluation in "dataset" in table 1 the same as the VPA training dataset?

---

> ### Author Response · Authors · 2024-11-20
> **response 1**
>
> Thank you for the insightful comments and constructive suggestions.
>
> **Q1: Concerns on VPA - pretrained Q functions**
>
> **A:** Thank you for the suggestion. We have the following response:
>
> 1) Although the pretrained Q functions can be inaccurate, they can still provide meaningful prior knowledge from the offline dataset and serve as an initial point for Q function finetuning.
> This would generally speed up the learning and lead to a better local optima compared to learning from scratch based on the offline data from a random initial point. Moreover, considering the limited number of steps allowed in VPA for efficiency, directly learning completely forgoes the knowledge learned offline and can fail to find good Q estimations.
>
> 2) As suggested by the reviewer, we add learning from scratch in VPA as a baseline and conduct additional experiments. As shown in Appendix D.3, we demonstrate the effectiveness of fine-tuning Q networks instead of training a new Q network. It is clear that the fine-tuning approach outperforms the training-from-scratch approach (using equations (8) and (9)). We will conduct the comparison on more benchmarks in the final revision.
>
> **Q2: Concerns on VPA - Distribution shift**
>
> **A:** This is a good point. However, the distribution shift only occurs during offline training, where the policy generating the data differs from the target policy. Once the offline-to-online transition begins, the proposed method operates fully in the online learning regime, where the data is collected under the updated online policy. As a result,
> the online policy update will not result in distribution shift, as the data aligns with the current policy. In our framework, the value Pre-Alignment (VPA), is implemented prior to online fine-tuning. A distribution shift occurs when the agent encounters out-of-distribution (OOD) states, leading to inaccurate Q-value estimation and suboptimal policy decisions in these scenarios.
> Our proposed VPA will mitigate Q estimation error from this aspect.
> As shown in Table 1 of the paper, when the initial state is randomly selected—falling outside the distribution of the offline dataset—the Q-values become more reasonable after applying VPA. This demonstrates that, from the perspective of mitigating inaccuracies in Q estimation due to distribution shifts in OOD states, our proposed VPA effectively alleviates this issue to some extent.
>
> **Q3: PID Lagrangian update**
>
> **A:** We have the following response to the reviewer's comments:
>
> 1) We kindly disagree with the reviewer that aPID only marginally improves over PID in Fig. 5. More specifically, aPID helps make the agent's cost more stable and lead to fewer constraint violations during the online fine-tuning process, which is very important for safe RL in practice.
> In all environments, aPID demonstrates superior stability in the cost training curve compared to PID. Moreover, in BallCircle, CarCircle, and HalfCheetah, aPID also achieves better performance in terms of reward, outperforming PID by approximately 100, 100, and 300, respectively.
> Therefore, the performance improvement of aPID over PID cannot be considered marginal.
>
> 2) Although our method aPID introduces more parameters, this is very common for adaptive algorithms in order to control the adaptation during the learning procedure. Besides,  the additional parameters are very easy to tune and the performance of aPID is robust to the selection of these parameters. To show this, we have conducted additional results and added them to Appendix D.4.2 in the revision. In the experiments, we evaluate the performance of our method under a wide range of these parameters, from 0.01 to 0.5. It is clear that the performance of our method is consistently good under different parameter selections.
>
> 3) We also kindly disagree with the reviewer that the parameters of PID are not well tuned in our experiments. First, because of the differences in the settings of training and environment, our selection of parameters for PID is different from that in [1-3], which has already been optimized. To justify this, we further conduct experiments by using the parameters of PID provided by the FSRL library, and add the results in Appendix D.4.1 in the revision.
> In the experiments,
> "Our PID" and "Our aPID" refer to using the PID and aPID parameters proposed in this paper for adjusting the Lagrange multipliers, respectively. Similarly, "FSRL PID" and "FSRL aPID" use the parameters  provided by the FSRL library.
> It is clear that the implementation of PID in our paper indeed significantly outperforms the implementation of PID provided by FSRL. More importantly, even with these suboptimal PID parameters, adaptive adjustment through aPID still leads to performance improvements and much stable cost (particularly important for safe RL).

---

> ### Author Response · Authors · 2024-11-20
> **response 2**
>
> **Q4: Experimental results**
>
> **A:** Thanks for your comments. We have the following clarifications about the experimental results:
>
> 1) As we mentioned in the paper, offline-to-online safe RL has been rarely explored in the literature, and the only existing method to study this is Guided Online Distillation (GOD), which uses Decision Transformer based on GPT-2 as the pretrained policy and follows JSRL for online finetuning. To ensure fair evaluations, we  compare with JSRL instead of GOD  without using GPT-2. There are no other baseline approaches we can compare, to the best of our knowledge.
> We will add VPA from scratch as an additional baseline as suggested by the reviewer, and will appreciate it if the reviewer can point out any validate baselines.
> In contrast, there are more studies for offline-to-online RL in the unconstrained case. To better justify the unique challenges in offline-to-online safe RL and the effectiveness of our framework, we adapt current SOTA offline-to-online RL approaches in the unconstrained case to the setup considered in our paper. Because these  offline-to-online RL algorithms were not specifically designed for constrained RL, it is understandable that directly applying these approaches would not achieve good performance, which in turn justifies the need of designing new frameworks to handle the unique challenges in offline-to-online RL. However, certain offline-to-online algorithms, e.g., SO2,   demonstrate strong performance in the CarRun environment. This is because SO2's design focuses on more accurate Q-value estimation, a principle that can be extended to offline-to-online safe RL. On the other hand, algorithms like JSRL and PEX, which rely on exploration policies, are less suited for safe RL scenarios.
>
> 2) In terms of the performance of purely online learning,  we think the reviewer may misunderstand the advantage/setup of offline-to-online RL and hope our following explanations can clarify this.
> The primary goal of offline-to-online algorithms is to achieve competitive performance with **minimal** environment interactions and in the **shortest time**, by leveraging offline information to speed up online learning. In contrast, the algorithm in [3] (almost the same for all the online algorithms) achieves higher performance but relies on **significantly more interactions**. For instance, in the BallCircle environment, [3] utilized 1.5 million interactions, whereas our method required only 120,000 interactions (with an average of 600 interactions per gradient update). This explains the performance difference relative to [3], and underscores the efficiency of our approach in resource-constrained and safety-critical settings where only a limited number of online interactions is allowed.
> We have also done additional experiments to compare our implementation of SAC-lag with that provided by the FSRL library, under the environment and training setting in our paper (including both environment interaction steps and policy update frequencies). This result is shown in Appendix D.6 in the revision. It can be seen that both implementations indeed demonstrate very similar performance in terms of both reward and cost, which is sufficient to verify the correctness of our implementation of SAC-lag.
>
> **Q5: tSNE**
>
> **A:** Thank you for pointing out the confusion regarding the input to t-SNE in Fig. 1(b). Below, we provide a detailed clarification:
>
> 1. Input to t-SNE:
> The input to t-SNE is the state vector corresponding to each state in the environment. These state vectors are high-dimensional representations derived from the environment. t-SNE reduces these high-dimensional state vectors to a 2D space for visualization purposes. Each point in Fig. 1(b) represents a state in this 2D space.
>
> 2. Sparse Cost Visualization Explanation:
> While every state has an associated reward, only a subset of states is associated with a non-zero cost. This sparsity is reflected in the visualization, as only a limited number of points (states) are highlighted with cost information. In contrast, all states contribute to the reward distribution. As a result, states associated with costs appear as sparse clusters or isolated points in the 2D visualization, whereas states with rewards are more uniformly distributed across the space.
>
> **Q6: Meaning of $\mu$ and $\hat{Q}$ in (8)**
>
> **A:** Sorry for the misunderstanding. $\mu$ refers to parameters of Q network. $\hat{Q}$ refers to target Q network.

---

> ### Author Response · Authors · 2024-11-20
> **response 3**
>
> **Q7: Table 1**
>
> **A:** The Q-values represent the expected cumulative reward from a given state when following a specific policy, while the Qc-values represent the expected cumulative cost. These values are estimated through Monte Carlo (MC) simulations, making them accurate because the simulations explicitly capture the sequential interactions of the agent with the environment under the given policy.
>
> For the MC simulations, we use the pre-trained policy derived from the training phase. Each simulation starts from a selected initial state. The number of interaction steps with the environment depends on the specific settings of the environment. For instance, in the BallCircle environment, the maximum number of steps is 200. A total of 10 Monte Carlo simulations are performed, and at each timestep, we record both the reward and the cost. To compute the true Q-values and Qc-values, the recorded rewards and costs are averaged cumulatively across all steps in the episodes.
>
> Regarding the choice of the initial state, the term "dataset" refers to selecting the initial state from the offline dataset used during VPA, whereas "random" indicates that the initial state is chosen randomly. This approach ensures a diverse evaluation and enhances the robustness of the estimated values.
>
> We provided clarification on this part in Appendix F.2 of the revised version of the paper.
>
> **end:**
>
> Finally, if our response resolves your concerns to a satisfactory level, we wonder if the reviewer could kindly consider raising the score of your evaluation. Certainly, we are more than happy to address any further questions that you may have during the discussion period. We thank the reviewer again for the helpful comments and suggestions for our work.

---

> ### Author Response · Authors · 2024-11-25
>
> Dear Reviewer,
>
> As the author-reviewer discussion period will end soon, we will appreciate it if you could check our response to your review comments. This way, if you have further questions and comments, we can still reply before the author-reviewer discussion period ends. If our response resolves your concerns, we kindly ask you to consider raising the rating of our work. Thank you very much for your time and efforts!

---

### Author Response · Authors · 2024-11-20
**general response**

We sincerely thank all the reviewers for their comments and thank the AC for handling this paper. We provide our responses to the reviewers' comments below and have made major revisions in our revised manuscript. Due to the page limit, we put our additional experiments in the appendix. To enhance clarity, we have highlighted the revised text in **blue** for easy identification.
Besides the specific response to each reviewer, we also list our response to some general questions.

**1. Purpose and Setup of Offline-to-Online Safe  RL:**

The primary goal of offline-to-online safe RL is to leverage offline data to facilitate efficient online fine-tuning, achieving competitive performance with minimal environment interactions. This is particularly important in safety-critical applications where interactions may be costly or risky.

In contrast to pure online safe RL, which requires a large number of interactions to learn from scratch, our approach starts with a policy and value functions pretrained on offline data. This warm-start allows the agent to quickly adapt and improve performance during online learning, reducing the need for extensive exploration that could lead to safety violations.

Moreover, compared to pure safe offline RL, which highly depends on the quality of offline data cannot improve beyond the limitations of the offline dataset, offline-to-online safe RL allows the agent to continue learning and improving by interacting with the environment, while still maintaining safety constraints.

Therefore, an important evaluation criteria for effective offline-to-online safe RL methods is whether the method can quickly obtain a good policy with high rewards and low safety violations only after a limited number of online interactions, which is the setup considered in this paper.

**2. Performance of Pure Online Algorithms such as SAC-Lag:**

Pure online algorithms like SAC-Lag can achieve high performance but require significantly more environment interactions. Our work emphasizes efficiency in terms of interaction cost and time. The key advantage of our offline-to-online framework is the ability to achieve strong performance with a limited number of online interactions, leveraging the offline pretrained policy and value functions to accelerate learning. This efficiency is crucial in practical settings where safety and resource constraints limit the amount of allowable interaction with the environment.

Some reviewers questioned the performance of our implementation of SAC-Lag compared to other implementations, suggesting that our SAC-Lag baseline might not be optimally tuned. We would like to clarify that our implementation of SAC-Lag is correct and comparable to standard implementations, such as those provided by the FSRL library.

To validate this, we conducted additional experiments using the SAC-Lag implementation from the FSRL library under the same environment settings, including the number of environment interactions and policy update frequencies, as in our experiments. Notably, in its default settings, the FSRL implementation utilized 1.5 million interactions to achieve a reward of approximately 700. In contrast, our method achieved comparable performance with only 120,000 interactions (with an average of 600 interactions per gradient update). To ensure fairness, we also evaluated a pure online SAC-Lag implementation with a limit of 120,000 interactions, finding that if the FSRL implementation of SAC-Lag used the same settings, its performance would be similar to ours.

These results confirm that our SAC-Lag baseline is correctly implemented and appropriately tuned, demonstrating its validity for comparisons in our study.

**3. Parameters in Adaptive PID (aPID) Controller:**

Although our method aPID introduces more parameters, this is very common for adaptive algorithms in order to control the adaptation during the learning procedure. And we have also  shown through additional  experiments that these parameters are easy to tune and the algorithm's performance is robust to their selection.

In our experiments, we used the same set of aPID parameters across all environments, demonstrating the method's generality. We also conducted sensitivity analyses, varying the aPID parameters over a wide range (from 0.01 to 0.5), and found that the performance remains consistently good. This indicates that the additional parameters do not impose a significant tuning burden and that aPID effectively improves stability and performance in safe RL settings.
***
We hope that these clarifications address the common concerns and highlight the contributions of our work. We are committed to further refining the manuscript and incorporating any additional feedback.

---

### Meta-Review · Area_Chair_4LMp · 2024-12-21

**Metareview:**

Summary:
This paper investigates safe reinforcement learning (RL) under the offline-to-online (O2O) paradigm. It introduces the MARVEL framework, which employs Value Pre-Alignment to align Q-functions with the underlying ground truth before transitioning to online learning, and Adaptive PID Control to dynamically adjust Lagrange multipliers during online fine-tuning.

Strengths:

- The focus on O2O in safe RL is an important yet underexplored topic.
- The paper systematically addresses this gap by analyzing the O2O problem, formulating it, and then proposing a concrete solution.
- The writing is clear, with a logical flow from problem analysis to solution, making it accessible for readers to follow.

Weaknesses:

- The experimental results are not sufficiently compelling. Certain experimental setups require additional detail to justify their choices.
- A more comprehensive demonstration of how the proposed method improves O2O efficiency would bolster the paper’s contributions.

Decision:
Reject.

**Additional Comments On Reviewer Discussion:**

During the discussion, the authors provided additional experiments evaluating the stability of the PID component in the MARVEL framework, as well as supplementary results for several baseline methods. However, since the primary objective of this submission is to enhance O2O efficiency, a more comprehensive set of experiments and analyses is still necessary to substantiate the claims.

---

### Decision · Program_Chairs · 2025-01-22

Reject